# Improving Intrinsic Exploration by Creating Stationary Objectives

**Roger Creus Castanyer**
Mila Québec AI Institute
Université de Montréal

**Joshua Romoff**
Ubisoft LaForge
joshua.romoff@ubisoft.com

**Glen Berseth**
Mila Québec AI Institute
Université de Montréal

{roger.creus-castanyer, glen.berseth}@mila.quebec

## Abstract

Exploration bonuses in reinforcement learning guide long-horizon exploration by defining custom intrinsic objectives. Several exploration objectives like count-based bonuses, pseudo-counts, and state-entropy maximization are non-stationary and hence are difficult to optimize for the agent. While this issue is generally known, it is usually omitted and solutions remain under-explored. The key contribution of our work lies in transforming the original non-stationary rewards into stationary rewards through an augmented state representation. For this purpose, we introduce the Stationary Objectives For Exploration (SOFE) framework. SOFE requires *identifying* sufficient statistics for different exploration bonuses and finding an *efficient* encoding of these statistics to use as input to a deep network. SOFE is based on proposing state augmentations that expand the state space but hold the promise of simplifying the optimization of the agent's objective. We show that SOFE improves the performance of several exploration objectives, including count-based bonuses, pseudo-counts, and state-entropy maximization. Moreover, SOFE outperforms prior methods that attempt to stabilize the optimization of intrinsic objectives. We demonstrate the efficacy of SOFE in hard-exploration problems, including sparse-reward tasks, pixel-based observations, 3D navigation, and procedurally generated environments.

## 1 Introduction

Intrinsic objectives have been widely used to improve exploration in reinforcement learning (RL), especially in sparse-reward and no-reward settings. In the case of Markov Decision Processes (MDPs) with a finite and small set of states, count-based exploration methods perform near-optimally when paired with tabular RL algorithms (Strehl & Littman, 2008; Kolter & Ng, 2009). Count-based methods keep track of the agent's frequency of state visits to derive an exploration bonus that can be used to encourage structured exploration. While much work has studied how to extend these methods to larger state spaces and continuous environments (Bellemare et al., 2016; Lobel et al., 2023; Tang et al., 2017), count-based methods introduce unstable learning dynamics that have not been thoroughly studied and can make it impossible for the agent to discover optimal policies. Specifically, any reward distribution that depends on the counts (i.e. the state-visitation frequencies) is non-stationary because the dynamics for the counts change as the agents generate new experiences, and the agent does not have access to the information needed to estimate these dynamics. In an MDP, the convergence of policies and value functions relies on the transition dynamics and the reward distribution being stationary (Sutton & Barto, 2018). The non-stationarity of count-based rewards induces a partially observable MDP (POMDP), as the dynamics of the reward distribution are unobserved by the agent. In a POMDP, there are no guarantees for an optimal Markovian (i.e. time-homogeneous) policy to exist (Alegre et al., 2021; Cheung et al., 2020; Lecarpentier & Rachelson, 2019). In general, optimal policies in POMDPs will require non-Markovian reasoning to adapt to the dynamics of the non-stationary rewards (Seyedsalehi et al., 2023). Despite this issue, count-based methods are usually paired with RL algorithms that are designed to converge to Markovian policies and hence might attain suboptimal performance. Previous research has either overlooked or attempted to address the non-stationarity issue in intrinsic rewards (Singh et al., 2010).

Some efforts to tackle this problem involve completely separating the exploration and exploitation policies (Schäfer et al., 2021; Whitney et al., 2021). However, these approaches add an additional layer of complexity to the RL loop and can introduce unstable learning dynamics. In this work, we introduce a framework to define stationary objectives for exploration (SOFE). SOFE provides an intuitive algorithmic modification to eliminate the non-stationarity of the intrinsic rewards, making the learning objective stable and stationary. With minimal complexity, SOFE enables both tractable and end-to-end training of a single policy on the combination of intrinsic and extrinsic rewards.

SOFE is described in Section 4 and consists of augmenting the original states of the POMDP by including the state-visitation frequencies or a representative embedding. SOFE proposes a state augmentation that effectively formulates the intrinsic reward distribution as a deterministic function of the state, at the cost of forcing the agent to operate over a larger set of states. We hypothesize that RL agents with parametrized policies are better at generalizing across bigger sets of states than at optimizing non-stationary rewards. We evaluate the empirical performance of SOFE in different exploration modalities and show that SOFE enables learning better exploration policies. We present SOFE as a method to solve the non-stationarity of count-based rewards. However, we show that SOFE provides orthogonal gains to other exploration objectives, including pseudo-counts and state-entropy maximization. Furthermore, our experiments in Section 5 show that SOFE is agnostic to the RL algorithm and robust in many challenging environment specifications, including large 3D navigation maps, procedurally generated environments, sparse reward tasks, pixel-based observations, and continuous action spaces. Videos of the trained agents and summarized findings can be found on our supplementary webpage[1].

## 2    RELATED WORK

**Exploration in RL** Exploration is a central challenge in RL. Classical exploration strategies explore in an aleatoric fashion. $\epsilon$-greedy (Sutton & Barto, 2018) samples random actions during training for the sake of exploration. Adding random structured noise in the action space (Lillicrap et al., 2015; Fujimoto et al., 2018) can enable exploration in continuous spaces. Maximum entropy RL provides a framework to find optimal policies that are as diverse as possible, and hence better explore the space of solutions (Haarnoja et al., 2018; Levine et al., 2020; Jain et al., 2024). For hard-exploration tasks, structured exploration has been studied through the lens of hierarchical RL (Gehring et al., 2021; Eysenbach et al., 2018). State-entropy maximization has been proposed to explore efficiently, in an attempt to learn policies that induce a uniform distribution over the state-visitation distribution (Seo et al., 2021; Lee et al., 2019; Zisselman et al., 2023). In MDPs with sparse reward distributions, exploration bonuses (i.e. intrinsic rewards) provide proxy objectives to the agents that can induce state-covering behaviors, hence allowing agents to find the sparse rewards. Count-based methods (Auer, 2002) derive an exploration bonus from state-visitation frequencies. Importantly, the inverse counts of a given state measure its novelty and hence provide a suitable objective to train exploratory agents. This property makes count-based exploration an appealing technique to drive structured exploration. However, count-based methods do not scale well to high-dimensional state spaces (Bellemare et al., 2016). Pseudo-counts provide a framework to generalize count-based methods to high-dimensional and partially observed environments (Tang et al., 2017; Lobel et al., 2023; Bellemare et al., 2016).

In modern deep RL applications, many popular methods enable exploration by defining exploration bonuses in high-dimensional state spaces (Laskin et al., 2021), and among them are curiosity-based (Pathak et al., 2017; Burda et al., 2018), data-based (Yarats et al., 2021) and skill-based (Eysenbach et al., 2018; Lee et al., 2019). Recently, elliptical bonuses have achieved great results in contextual MDPs with high-dimensional states (Henaff et al., 2022). These methods aim to estimate novelty in the absence of the true state-visitation frequencies. Henaff et al. (2022) showed that elliptical bonuses provide the natural generalization of count-based methods to high-dimensional observations. In this work, we show that SOFE improves the performance of count-based methods in small MDPs and pseudo-counts in environments with high-dimensional observations (e.g. images), further improving the performance of the state-of-the-art exploration algorithm E3B in contextual MDPs. Additionally, our results show that SOFE provides orthogonal gains to exploration objectives of different natures like state-entropy maximization.

---

[1]https://sites.google.com/view/sofe-webpage/home

**Non-stationary objectives** A constantly changing (i.e. non-stationary) MDP induces a partially observed MDP (POMDP) if the dynamics of the MDP are unobserved by the agent. In Multi-Agent RL, both the transition and reward functions are non-stationary because these are a function of other learning agents that evolve over time (Zhang et al., 2021a; Papoudakis et al., 2019). In contextual MDPs, the transition and reward functions can change every episode and hence require significantly better generalization capabilities, which might not emerge naturally during training (Cobbe et al., 2020; Henaff et al., 2022; Wang et al., 2022). For MDPs with non-stationary rewards, meta-learning and continual learning study adaptive algorithms that can adapt to moving objectives (Beck et al., 2023). Learning separate value functions for non-stationary rewards has also been proposed (Burda et al., 2018). Schäfer et al. (2021) proposed DeRL, which entirely decouples the training process of an exploratory policy from the exploitation policy. While DeRL mitigates the effect of the non-stationary intrinsic rewards in the exploitation policy, the exploration policy still faces a hard optimization problem. Importantly, there might not exist an optimal Markovian policy for a POMDP (Seyedsalehi et al., 2023). Hence, RL algorithms can only achieve suboptimal performance in these settings.

Many exploration bonuses are non-stationary by definition. In particular, count-based methods are non-stationary since the state-visitation frequencies change during training (Singh et al., 2010; Şimşek & Barto, 2006). We note that this issue is also present in many of the popular deep exploration methods that use an auxiliary model to compute the intrinsic rewards like ICM (Pathak et al., 2017), RND (Burda et al., 2018), E3B (Henaff et al., 2022), density models (Bellemare et al., 2016; Tang et al., 2017) and many others (Lobel et al., 2023; Raileanu & Rocktäschel, 2020; Flet-Berliac et al., 2021; Zhang et al., 2021b). In these cases, the non-stationarity is caused by the weights of the auxiliary models also changing during training. In this work, we argue that non-stationarity should not be implicit when an exploration bonus is defined. For this reason, we introduce SOFE, which proposes an intuitive modification to intrinsic objectives that eliminates their non-stationarity and facilitates the optimization process.

## 3 PRELIMINARIES

Reinforcement Learning (RL) uses MDPs to model the interactions between a learning agent and an environment. An MDP is defined as a tuple $\mathcal{M} = (\mathcal{S}, \mathcal{A}, \mathcal{R}, \mathcal{T}, \gamma)$ where $\mathcal{S}$ is the state space, $\mathcal{A}$ is the action-space, $\mathcal{R} : \mathcal{S} \times \mathcal{A} \rightarrow \mathbb{R}$ is the extrinsic reward function, $\mathcal{T} : \mathcal{S} \times \mathcal{A} \times \mathcal{S} \rightarrow [0, 1]$ is a transition function and $\gamma$ is the discount factor. The objective of the agent is to learn a policy that maximizes the expected discounted sum of rewards across all possible trajectories induced by the policy. If the MDP is non-stationary, then there exists some unobserved environment state that determines the dynamics of the MDP, hence inducing a partially observed MDP (POMDP), which is also a tuple $\mathcal{M}' = (\mathcal{S}, \mathcal{O}, \mathcal{A}, \mathcal{R}, \mathcal{T}, \gamma)$ where $\mathcal{O}$ is the observation space and the true states $s \in \mathcal{S}$ are unobserved. In a POMDP, the transition and reward functions might not be Markovian with respect to the observations, and therefore, the policy training methods may not converge to an optimal policy. To illustrate this, consider an MDP where the reward distribution is different at odd and even time steps. If the states of the MDP are not augmented with an odd/even component, the rewards appear to be non-stationary to an agent with a Markovian policy. In this case, a Markovian policy will not be optimal over all policies. The optimal policy will have to switch at odd/even time steps. In this work, we extend the previous argument to intrinsic exploration objectives in RL. In the following sections, we uncover the implicit non-stationarity of several exploration objectives and propose a novel method to resolve it.

### 3.1 EXPLORATION BONUSES AND INTRINSIC REWARDS

In hard-exploration problems, exploration is more successful if directed, controlled, and efficient. Exploration bonuses provide a framework to decouple the original task from the exploration one and define exploration as a separate RL problem. In this framework, the extrinsic rewards provided by the environment are aggregated with the intrinsic rewards (i.e. exploration bonuses) to build an augmented learning target. By directing the agent's behavior towards custom exploration bonuses, this formulation induces exploratory behaviors that are state-covering and are well-suited for long-horizon problems.

Central to SOFE is the introduction of the parameters $\phi_t$ in the formulation of exploration bonuses $\mathcal{B}(s_t, a_t|\phi_t)$, which enables reasoning about the dynamics of the intrinsic reward distributions. The parameters of the intrinsic reward distribution $\phi_t$ determine how novelty is estimated and exploration is guided, and if they change over time then $\mathcal{B}$ is non-stationary. In the following, we unify count-based methods, pseudo-counts, and state-entropy maximization under the same formulation, which includes $\phi_t$. In the next section, we present SOFE as a solution to their non-stationarity.

### 3.1.1 COUNT-BASED METHODS

Count-based methods keep track of the agent's frequencies of state visits to derive an exploration bonus. Formally, the counts keep track of the visited states until time $t$, and so $\mathcal{N}_t(s)$ is equal to the number of times the state $s$ has been visited by the agent until time $t$. Two popular intrinsic reward distributions derived from counts that exist in prior work are:

$$\mathcal{R}(s_t, a_t, s_{t+1}|\phi_t) = \mathcal{B}(s_t, a_t, s_{t+1}|\phi_t) = \frac{\beta}{\sqrt{\mathcal{N}_t(s_{t+1}|\phi_t)}} \tag{1}$$

where $\beta$ weights the importance of the count-based bonus, and:

$$\mathcal{R}(s_t, a_t, s_{t+1}|\phi_t) = \mathcal{B}(s_t, a_t, s_{t+1}|\phi_t) = \left\{ \begin{array}{ll} 1, & \text{if } \mathcal{N}_t(s_{t+1}|\phi_t) = 0 \\ 0, & \text{else} \end{array} \right. \tag{2}$$

Note that the state-visitation frequencies $\mathcal{N}_t$ are the sufficient statistics for $\phi_t$ and hence for the count-based rewards in Equations 1 and 2. That is, the state-visitation frequencies are the only dynamically changing component that induces non-stationarity in count-based rewards.

Equation 1 (Strehl & Littman, 2008) produces a dense learning signal since $\mathcal{B}(s_t, a_t, s_{t+1}|\phi_t) \neq 0$ unless $\mathcal{N}_t(s_{t+1}) = \infty$ which is unrealistic in practice. Equation 2 (Henaff et al., 2023) defines a sparse distribution where the agent is only rewarded the first time it sees each state, similar to the objective of the travelling salesman problem. Throughout the paper, we refer to Equations 1 and 2 as $\sqrt{}$-*reward* and *salesman reward*.

### 3.1.2 PSEUDO-COUNTS

To enable count-based exploration in high-dimensional spaces, the notion of visitation counts has been generalized to that of pseudo-counts (Bellemare et al., 2016). Prior work has estimated pseudo-counts through density models (Bellemare et al., 2016), neural networks (Ostrovski et al., 2017), successor representations (Machado et al., 2020), and samples from the Rademacher distribution (Lobel et al., 2023). Recently, Henaff et al. (2022) proposed elliptical bonuses (E3B) as a natural generalization of count-based methods. An appealing property of E3B is that it models the complete set of state-visitation frequencies over the state space, and not only for the most recent state[2] Concretely, the E3B algorithm produces a bonus:

$$\mathcal{B}(s_t, a_t, s_{t+1}|\phi_t) = \psi_t(s_{t+1})^T C_t^{-1} \psi_t(s_{t+1})$$
$$C_t = \sum_{t=0}^{T} \psi_t(s_t)\psi_t(s_t)^T \tag{3}$$

where $\psi_t$ is an auxiliary model that produces low-dimensional embeddings from high-dimensional observations. Since the ellipsoid is updated after each transition, the exploration bonus is non-stationary. The matrix $C_t$ defines an ellipsoid in the embedding space, which encodes the distribution of observed embeddings in a given trajectory. Since $C_t$ is the only moving component of Equation 3, it is a sufficient statistic to characterize the non-stationarity of the reward distribution. Note that in an MDP with finite state space, where $\psi$ is the one-hot encoding of the states, the exploration bonus in Equation 3 becomes a count-based bonus similar to Equation 1. Concretely, $C_{t-1}^{-1}$ becomes a diagonal matrix with the inverse state-visitation frequencies for each state in the elements of the diagonal (Henaff et al., 2022).

---

[2]Previous pseudo-count methods allowed the agent to query a density model with a single state and obtain its pseudo-count. However, E3B maintains a model of the state-visitation frequencies over the complete state space. The latter is key for SOFE to obtain sufficient statistics of the E3B reward in Equation 3.

### 3.1.3 STATE-ENTROPY MAXIMIZATION

State-entropy maximization is a widely used exploration objective that consists of training policies to induce a uniform distribution over the state-marginal visitation distribution (Lee et al., 2019). A canonical formulation of this problem is presented in Berseth et al. (2019). Maximizing the state-entropy objective corresponds to training policies to maximize the following reward distribution (Berseth et al., 2019):

$$\mathcal{R}(s_t, a_t, s_{t+1}|\phi_t) = \mathcal{B}(s_t, a_t, s_{t+1}|\phi_t) = -\log p_{\phi_t}(s_{t+1}) \tag{4}$$

The parameters $\theta$ define the policy and $\phi$ are the parameters of a generative model which estimates the state-marginal distribution $d^{\pi_\theta}(s_t)$. Note that the sufficient statistics of the generative distribution are also sufficient statistics for the intrinsic reward in Equation 4. Throughout the paper, we refer to this algorithm as *surprise maximization* (S-Max) and use the Gaussian distribution to model trajectories of states with $p_{\phi_t}$. Hence the sufficient statistics of $p_{\phi_t}$ for the reward reward in Equation 4 are $\phi_t = (\mu_t \cup \sigma_t^2)$. We present the details of S-Max in Section A.7.

## 4 STATIONARY OBJECTIVES FOR EXPLORATION

In the following, we present a training framework for Stationary Objectives for Exploration (SOFE). Any exploration bonus $\mathcal{B}(s_t, a_t, s_{t+1}|\phi_t)$ derived from dynamically changing parameters will define a non-stationary reward function. Without any modification, exploration bonuses define a POMDP: $\mathcal{M} = (\mathcal{S}, \mathcal{O}, \mathcal{A}, \mathcal{B}, \mathcal{T}, \gamma)$. For simplicity, we have fully replaced the task-reward $\mathcal{R}$ with the exploration bonus $\mathcal{B}$, and we consider that the only unobserved components in the POMDP are the parameters of the reward distribution[3]. Hence, we argue that the unobserved states $s \in S$ satisfy $s_t = o_t \cup \phi_t$. Note that the transition function of the POMDP is generally only Markovian if defined over the state space and not over the observation space: $\mathcal{T} : \mathcal{S} \times \mathcal{A} \times \mathcal{S} \to [0, 1]$.

The sufficient statistics for exploration bonuses are always available during training as they are explicitly computed to produce the intrinsic rewards. However, current RL methods do not allow the agents to observe them. Hence, any method that aims to solve $\mathcal{M}$ faces optimizing a non-stationary objective, which is difficult to optimize, as it can require non-Markovian properties like memory, continual learning, and adaptation, and may only find suboptimal policies. In this work, we argue that non-stationarity should not be implicit in the formulation of an exploration objective. For this reason, we propose SOFE, which augments

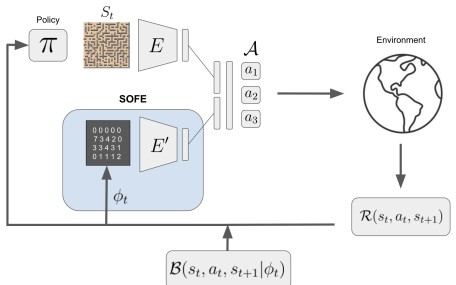

Figure 1: SOFE enables agents to observe the sufficient statistics of the intrinsic rewards and use them for decision-making.

the state space by defining an augmented MDP $\hat{\mathcal{M}} = \left(\hat{\mathcal{S}}, \mathcal{A}, \mathcal{B}, \mathcal{T}, \gamma\right)$ where $\hat{\mathcal{S}} = \{\mathcal{O} \cup \phi\}$, with $\mathcal{O}$ being the observations from $\mathcal{M}$. Note that we get rid of the observation space $\mathcal{O}$ in the definition of $\hat{\mathcal{M}}$ because by augmenting the original observations from $\mathcal{M}$ with the sufficient statistics for $\mathcal{B}$ we effectively define a fully observed MDP. This simple modification allows instantiating the same exploration problem in a stationary and Markovian setting. That is the optimal policies in $\hat{\mathcal{M}}$ are also optimal in $\mathcal{M}$. This is true since the transition and reward functions are identical in $\mathcal{M}$ and $\hat{\mathcal{M}}$. We note that the update rule for the parameters $\phi_t$ must be Markovian, meaning that these can be updated after every step without requiring information other than $s_t$ and $s_{t+1}$. For example, counts only increment by one for the state that was most recently visited: $\mathcal{N}_{t+1}(s) = \mathcal{N}_t(s) \forall s \in \{S - s_j\}$, where $s_j = s_{t+1}$ and $\mathcal{N}_{t+1}(s_j) = \mathcal{N}_t(s_j) + 1$. The latter also applies to E3B and S-Max, since the ellipsoid $C_t$ and parameters of the generative model are updated incrementally with every new transition (see Equation 3 and Section A.7). Given the sufficient statistics, the intrinsic reward distributions in Equations 1,2, 3, 4 become fully Markovian, and hence are invariant across time.

---

[3]This assumption holds true if the agent has access to sufficient statistics of the transition dynamics (e.g. grid environments), and makes SOFE transform a POMDP into a fully observed MDP. Even when there are unobserved components of the true states apart from the parameters of the intrinsic reward distributions, we empirically show that SOFE mitigates the non-stationary optimization, yielding performance gains.

## 5 EXPERIMENTS

SOFE is designed to improve the performance of exploration tasks. To evaluate its efficacy, we study three questions: (1) How much does SOFE facilitate the optimization of non-stationary exploration bonuses? (2) Does this increased stationarity improve exploration for downstream tasks? (3) How well does SOFE scale to image-based state inputs where approximations are needed to estimate state-visitation frequencies?

To answer each of these research questions, we run the experiments as follows. (1) We use three different mazes without goals to investigate how SOFE compares to vanilla count-based methods and S-Max in reward-free exploration. Concretely, we evaluate whether SOFE allows for better optimization of purely exploratory behaviors. We also use a large 3D environment with continuous state and action spaces which introduces complex challenges as it requires more than purely navigation skills.

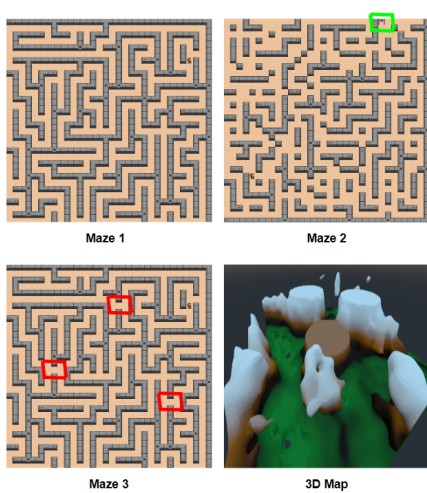

Figure 2: We use 3 mazes and a large 3D map to evaluate both goal-reaching and purely exploratory behaviors. Maze 1: a fully connected, hard-exploration maze; Maze 2: a maze with open spaces and a goal; Maze 3: same as Maze 1 but with 3 doors which an intelligent agent should use for more efficient exploration; 3D map: a large map with continuous state and action spaces.

Secondly (2), we use a 2D maze with a goal and sparse extrinsic reward distribution. This is a hard-exploration task where the extrinsic reward is only non-zero if the agent reaches the goal, which requires a sequence of 75 coordinated actions. We evaluate whether SOFE enables better optimization of the joint objective of intrinsic and task rewards. Furthermore, we use the DeepSea sparse-reward hard-exploration task from the DeepMind suite (Osband et al., 2019) and show that SOFE achieves better performance than DeRL (Schäfer et al., 2021) which attempts to stabilize intrinsic rewards by training decoupled exploration and exploitation policies.

Thirdly (3), we apply SOFE on the E3B (Henaff et al., 2022) algorithm as argued in Section 4 to demonstrate the effectiveness of the approach with an imperfect representation of the state-visitation frequencies. We use the *MiniHack-MultiRoom-N6-v0* task, originally used for E3B in Henaff et al. (2023), and the *Procgen-Maze* task (Cobbe et al., 2020). In both environments, the task is to navigate to the goal location in a procedurally generated map and the extrinsic reward is only non-zero if the agent reaches the goal. Both environments return pixel observations. Minihack additionally returns natural language observations. However, the *Procgen-Maze* task is more challenging because each episode uses unique visual assets, requiring an additional level of generalization, while in Minihack, different episodes only vary in the map layout. Additionally, we include the Habitat environment (Szot et al., 2021) to evaluate purely exploratory behaviors and show the results in Section A.1.

We provide the details of the network architectures, algorithm hyperparameters, and environment specifications in Section A.3. Furthermore, we provide an in-depth analysis of the behaviors learned by SOFE in Section A.2, which uncovers valuable insights on how SOFE learns to drive exploration more efficiently.

### 5.1 REWARD-FREE EXPLORATION

In this section, we focus on the first research question and consider the reward-free setting to evaluate purely exploratory behaviors. We use the 3 mazes in Figure 2 and measure map coverage, which correlates with exploration in navigation environments. In Figure 3, we show how SOFE enables agents to explore the mazes better than vanilla count-based methods. Even though we fix the count-based rewards described in Equations 1 and 2, SOFE generally enables RL agents to better optimize them, achieving higher state coverage. Section A.9 contains the results across all algorithms

and exploration modalities. We also run experiments in a large 3D environment from the Godot

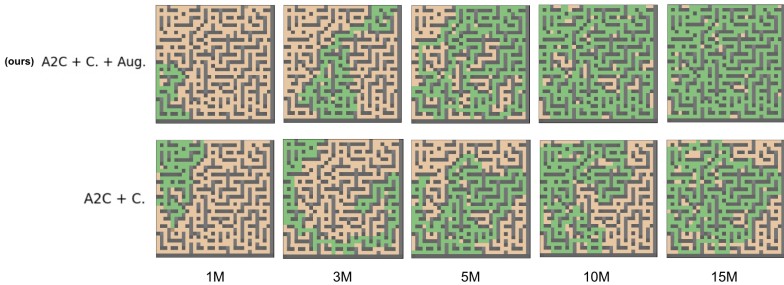

Figure 3: Episodic state-visitation for A2C agents during training. The first row represents SOFE, which uses both the count-based rewards and state augmentation (+ C. + Aug.), and the second row represents training with the count-based rewards only (+ C.). Although optimizing for the same reward distribution, our method achieves better exploration performance.

RL repository (Beeching et al., 2021a), to evaluate SOFE's ability to scale to continuous state and action spaces. This environment contains challenging dynamics that require exploratory agents to master a variety of skills, from avoiding lava and water to using jump pads efficiently (Alonso et al., 2020; Beeching et al., 2021b). Figure 4 shows that SOFE also scales to these more complex settings, enabling SAC (Haarnoja et al., 2018) agents to achieve higher map coverage across different exploration modalities.

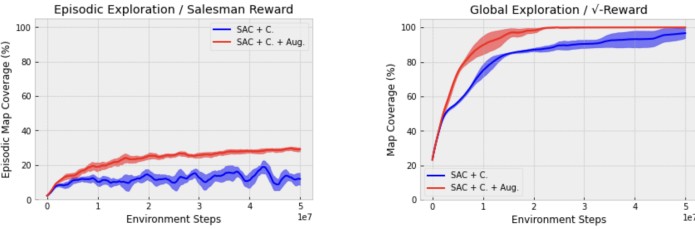

Figure 4: Map coverage achieved by SAC agents in a complex 3D map. Blue curves represent agents that use count-based rewards (+ C.); Red curves represent SOFE, which uses both count-based rewards and the state augmentations from SOFE (+ C. + Aug.). Even though we use the same learning objective, SOFE facilitates its optimization and achieves better exploration. Shaded areas represent one standard deviation. Results are averaged from 6 seeds.

Additionally, we show that SOFE stabilizes the state-entropy maximization objective. Figure 5 shows the episodic map coverage achieved in *Maze 2* by the vanilla S-Max algorithm compared to the augmented S-Max with SOFE. These results provide further evidence that SOFE is a general framework that tackles the non-stationarity of exploration objectives and provides orthogonal gains across objectives of different natures.

## 5.2 EXPLORATION FOR SPARSE REWARDS

In the previous section, we showed that SOFE enables RL agents to better explore the state space. In this section, we evaluate whether SOFE can achieve better performance on hard-exploration tasks.

We evaluate count-based methods and SOFE in Maze 2 in Figure 2. For each of the RL algorithms, we compare training with the sparse extrinsic reward only and training with the extrinsic and intrinsic rewards with and without SOFE. Figure 6 shows that SOFE significantly improves the performance of RL agents in this hard-exploration task. Our results confirm that extrinsic rewards are not enough to solve such hard-exploration tasks and show that SOFE is significantly more effective than vanilla count-based methods, achieving the highest returns across multiple RL algorithms. PPO (Schulman et al., 2017), PPO+LSTM (Cobbe et al., 2020), and A2C (Mnih et al., 2016) achieve near-optimal

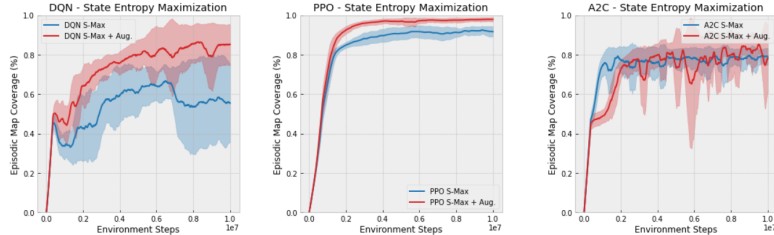

Figure 5: Episodic state coverage achieved by S-Max (blue) and SOFE S-Max (red) in Maze 2. When augmented with SOFE, agents better optimize for the state-entropy maximization objective. Shaded areas represent one standard deviation. Results are averaged from 6 seeds.

goal-reaching performance only when using SOFE. Importantly, policies equipped with LSTMs have enough capacity to model the non-stationary intrinsic rewards (Ni et al., 2021) and could learn to count implicitly (Suzgun et al., 2018). However, our results show that SOFE further improves the performance of recurrent policies when optimizing for non-stationary intrinsic rewards. Addi-

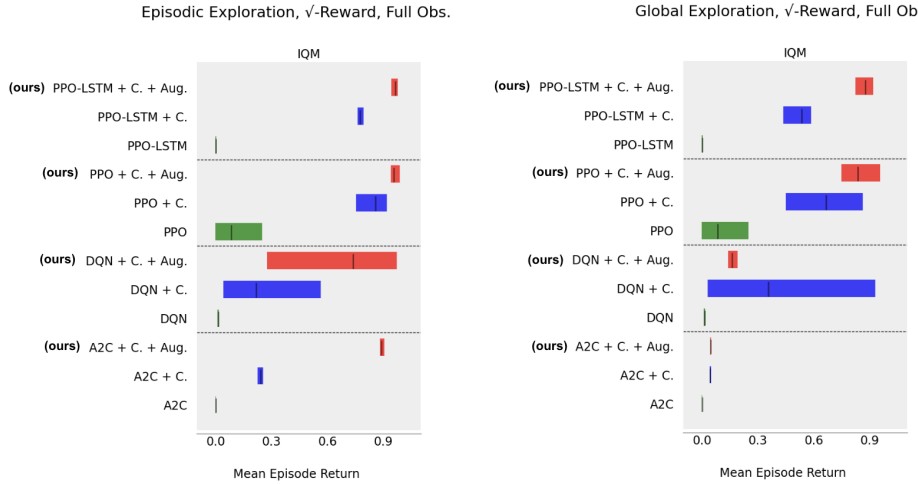

Figure 6: Interquantile mean (IQM) of episode extrinsic rewards for episodic (left) and global (right) exploration across multiple algorithms. Green bars represent training with the sparse reward; blue bars represent additionally using count-based rewards (+ C.); and red bars represent additionally using count-based rewards and SOFE (+ C. + Aug.). We compute confidence intervals using stratified bootstrapping with 6 seeds. SOFE generally achieves significantly higher extrinsic rewards across RL algorithms and exploration modalities. Without exploration bonuses, agents fail to obtain a non-zero extrinsic reward during training.

tionally, we compare SOFE to DeRL (Schäfer et al., 2021) in the DeepSea environment and show the results in Table 1. DeRL entirely decouples the training process of an exploratory policy from the exploitation policy to stabilize the optimization of the exploration objective. SOFE is degrees of magnitude less complex than DeRL as it only requires training an additional feature extractor. Still, SOFE achieves better results in the harder variations of the DeepSea environment.

## 5.3 EXPLORATION IN HIGH-DIMENSIONAL ENVIRONMENTS

In this section, we evaluate SOFE and E3B (Henaff et al., 2022), the state-of-the-art exploration algorithm for high-dimensional contextual MDPs. E3B tackles the challenging problem of estimating the true state-visitation frequencies from pixel observations. As argued in Section 5.3, the ellipsoid is the only moving component of the E3B objective. Hence, we evaluate whether including either the diagonal or the full ellipsoid in the state enables better exploration. We optimize the E3B objec-

| Algorithm | DeepSea 10 | DeepSea 14 | DeepSea 20 | DeepSea 24 | DeepSea 30 |
|-----------|-----------|-----------|-----------|-----------|-----------|
| DeRL-A2C | **0.98 ± 0.10** | 0.65 ± 0.23 | 0.42 ± 0.16 | 0.07 ± 0.10 | 0.09 ± 0.08 |
| DeRL-PPO | 0.61 ± 0.20 | 0.92 ± 0.18 | −0.01 ± 0.01 | 0.63 ± 0.27 | −0.01 ± 0.01 |
| DeRL-DQN | **0.98 ± 0.09** | **0.95 ± 0.17** | 0.40 ± 0.08 | 0.53 ± 0.27 | 0.10 ± 0.10 |
| SOFE-A2C | 0.94 ± 0.19 | 0.45 ± 0.31 | 0.11 ± 0.25 | 0.08 ± 0.14 | 0.04 ± 0.09 |
| SOFE-PPO | 0.77 ± 0.29 | 0.67 ± 0.33 | 0.13 ± 0.09 | 0.07 ± 0.15 | 0.09 ± 0.23 |
| SOFE-DQN | **0.97 ± 0.29** | 0.78 ± 0.21 | **0.70 ± 0.28** | **0.65 ± 0.26** | **0.42 ± 0.33** |

Table 1: Average performance of DeRL and SOFE with one standard deviation over 100,000 evaluation episodes in the DeepSea environment. SOFE achieves better performance in the harder exploration variations of the DeepSea environment while consisting of a less complex method. Additional details on the hyperparameters used for SOFE and DeRL are presented in Section A.3.

tive with IMPALA (Espeholt et al., 2018) as proposed in Henaff et al. (2022). Section A.3 contains the details of the policy architecture.

Figure 7 shows that SOFE also improves the performance of pseudo-count-based methods, providing empirical evidence that reducing the non-stationarity of a reward distribution enables better optimization even in high-dimensional environments. In Section A.1, we include experiments with E3B and SOFE in the reward-free setting using the Habitat simulator. These show that SOFE improves the sample efficiency of E3B.

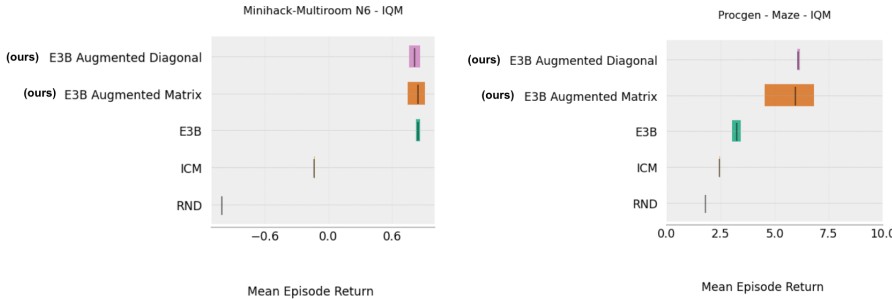

Figure 7: Interquantile mean (IQM) of the episode extrinsic rewards in *Minihack* (left) and *Procgen* (right). The vanilla E3B algorithm can solve MiniHack as shown in Henaff et al. (2022). However, we find that E3B can only consistently reach the goals in Procgen-Maze when using SOFE. As shown in Henaff et al. (2022), ICM (Pathak et al., 2017) and RND (Burda et al., 2018) fail to provide a good exploration signal in procedurally generated environments. Results are averaged from 6 seeds.

## 6 CONCLUSION

We identify that exploration bonuses can be non-stationary by definition, which can complicate their optimization, resulting in suboptimal policies. To address this issue, we have introduced a novel framework that creates stationary objectives for exploration (SOFE). SOFE is based on capturing sufficient statistics of the intrinsic reward distribution and augmenting the MDP's state representation with these statistics. This augmentation transforms the non-stationary rewards into stationary rewards, simplifying the optimization of the agent's objective. We have identified sufficient statistics of count-based methods, the state-entropy maximization objective, and E3B. Our experiments provide compelling evidence of the efficacy of SOFE across various environments, tasks, and reinforcement learning algorithms, even improving the performance of the state-of-the-art exploration algorithm in procedurally generated environments. Using augmented representations, SOFE significantly improves exploration behaviors, particularly in challenging tasks with sparse rewards and across multiple exploration modalities. Additionally, SOFE extends to large continuous state and action spaces, showcasing its versatility.

ACKNOWLEDGEMENTS

We want to acknowledge funding support from NSERC and CIFAR and compute support from Digital Research Alliance of Canada, Mila IDT and NVidia. We thank Mehran Shakerinava and Raj Ghugare for helping review the paper before submission.

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

# A  APPENDIX

## A.1  REWARD-FREE EXPLORATION WITH SOFE AND E3B

As in Section 5.1, we evaluate if SOFE can enable better optimization of the non-stationary exploration bonus, in this case for E3B. We consider the reward-free setting for purely exploratory behaviors. For this reason, we use the Habitat simulator (Savva et al., 2019; Szot et al., 2021) and the HM3D dataset (Ramakrishnan et al., 2021), which contains 1,000 different scenes of photorealistic apartments for 3D navigation. We train E3B and our proposed augmented versions for 10M environment steps and measure their map coverage in a set of 100 held-out scenes. We optimize the E3B exploration bonus with PPO (Schulman et al., 2017) which requires 31 hours in a machine with a single GPU. We show the results in Figure 8.

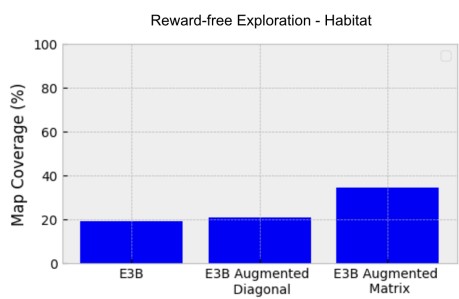

Figure 8: Map coverage on a held-out set of 100 3D scenes of the HM3D dataset. The E3B agents trained using SOFE explore the new scenes better.

In Figure A.1 we show the learning curves corresponding to the results presented in Section 5.3.

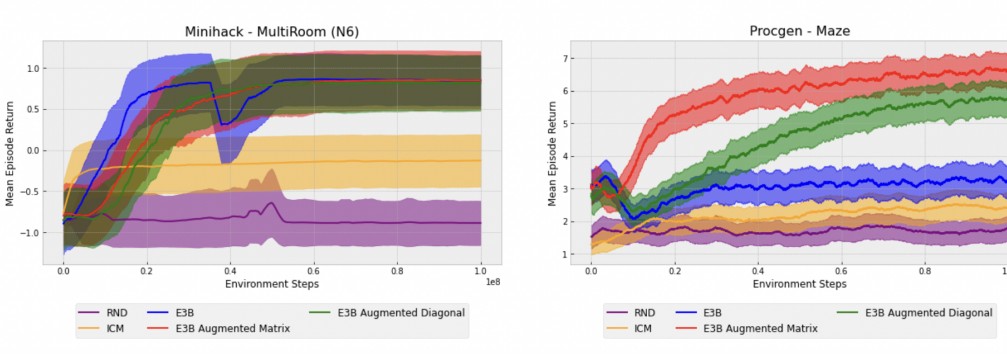

Figure 9: E3B (blue) outperforms the other deep exploration baselines in the hard-exploration and partially-observable tasks of MiniHack and Procgen-Maze. With SOFE (Red and Green), we further increase the performance of E3B.

## A.2    Analysis of the behaviours learned by SOFE

By using SOFE on count-based methods, RL agents extract features from the state-visitation frequencies and use them for decision-making. To better understand how the agents use the augmented information, we artificially create an object $\mathcal{N}_0$ with $\mathcal{N}_0(s_i) > 0 \; \forall_i \in \{\mathcal{S} - s_j\}$ and $\mathcal{N}_0(s_j) = 0$. Intuitively, we communicate to the agent that all states in the state space but $s_j$ have already been visited through the state-visitation frequencies. We evaluate PPO agents pre-trained on reward-free episodic exploration and show the results in Figure 10. Results show that augmented agents efficiently direct their exploration towards the unvisited states, self-identifying these as goals. This reveals how the agents leverage the augmented information for more efficient exploration.

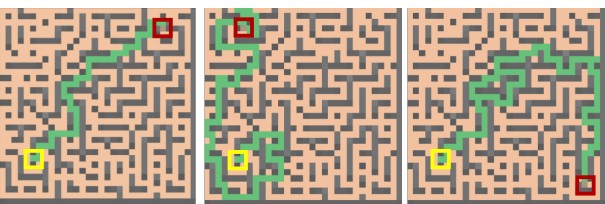

Figure 10: Analysis of how an RL agent uses SOFE for better exploration. Yellow boxes show the agent's starting position. Red boxes show the goals' positions, which are the only positions for which we set $\mathcal{N}_0(s_j) = 0$, and green traces show the agent's trajectory. Augmented agents pre-trained on episodic exploration efficiently direct their exploration toward the unvisited states.

## A.3    Training Details

### A.3.1    Network Architecture

We use Stable-Baselines3 (Raffin et al., 2021) to run our experiments in the mazes, Godot maps, and DeepSea. For DQN, PPO, A2C, and SAC we use the same CNN to extract features from the observation. The CNN contains 3 convolutional layers with kernel size of $(3 \times 3)$, stride of 2, padding of 1, and 64 channels. The convolutional layers are followed by a fully connected layer that produces observation embeddings of dimension 512. For the augmented agents, we use an additional CNN with the same configuration to extract features from $\phi_t$. The augmented agents concatenate the representations from the observation and the parameters $\phi_t$ and feed these to the policy for decision-making, while the vanilla methods (e.g. counts, S-Max) only extract features from the observations. We show the CNN architecture in Figure A.3.1.

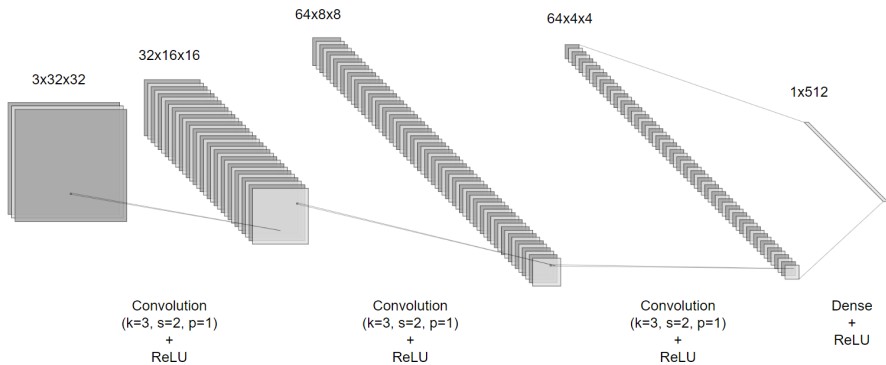

For Minihack and Procgen, we use the official E3B codebase, which contains baselines for ICM and RND, and uses IMPALA to optimize the exploration bonuses. We use the same policy architecture as in Henaff et al. (2022), which contains an LSTM. We ran the experiments in Minihack and Procgen for 100M steps. For the augmented agents, we design a CNN that contains 5 convolutional layers with a kernel size of $(3 \times 3)$ and stride and padding of 1, batch normalization layers after every convolutional layer, max-pooling layers with a kernel size of $(2 \times 2)$ and stride of 1, followed by

a fully-connected layer that produces embeddings of dimension 1024. This architecture allows to extract features from the 512x512 ellipsoids, which are later passed together with the observation features to the policy for decision-making. We show the CNN architecture in Figure **??**.

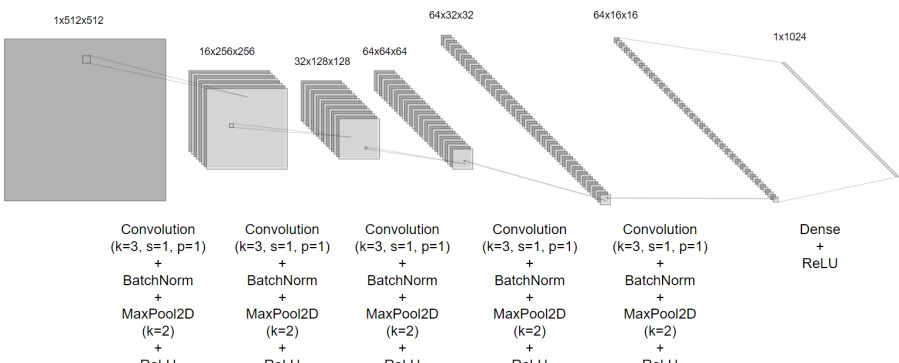

## A.4 ALGORITHM HYPERPARAMETERS

### A.4.1 DQN

| Hyperparameter | Value |
|---|---|
| Num. Envs | 16 |
| Learning Rate | 0.0001 |
| Buffer Size | 1000000 |
| Learning Starts | 50000 |
| Batch Size | 32 |
| Tau | 1.0 |
| Gamma | 0.99 |
| Train Frequency | 4 |
| Gradient Steps | 4 |
| Target Update Interval | 10000 |
| Exploration Fraction | 0.1 |
| Exploration Initial Epsilon | 1.0 |
| Exploration Final Epsilon | 0.05 |
| Max Grad Norm | 10 |

Table 2: Hyperparameters for the DQN Implementation

### A.4.2 PPO

| Hyperparameter | Value |
|---|---|
| Num. Envs | 16 |
| Learning Rate | 0.0003 |
| N Steps | 2048 |
| Batch Size | 64 |
| N Epochs | 10 |
| Gamma | 0.99 |
| GAE Lambda | 0.95 |
| Clip Range | 0.2 |
| Normalize Advantage | True |
| Entropy Coefficient | 0.0 |
| Value Function Coefficient | 0.5 |
| Max Grad Norm | 0.5 |

Table 3: Hyperparameters for the PPO Implementation

### A.5 A2C

| Hyperparameter | Value |
|---|---|
| Num. Envs | 16 |
| Learning Rate | 0.0007 |
| N Steps | 5 |
| Gamma | 0.99 |
| GAE Lambda | 1.0 |
| Entropy Coefficient | 0.0 |
| Value Function Coefficient | 0.5 |
| Max Grad Norm | 0.5 |
| RMS Prop Epsilon | $1 \times 10^{-5}$ |
| Use RMS Prop | True |
| Normalize Advantage | False |

Table 4: Hyperparameters for the A2C Implementation

## A.6 ENVIRONMENT DETAILS

### A.6.1 MAZES

We designed the mazes in Figure 2 with Griddly (Bamford, 2021). The 3 mazes are of size 32x32. The agents observe *entity maps*: matrices of size $(map\_size, map\_size)$ of entity id's (e.g. 0 for the floor, 1 for the wall, and 2 for the agent). The action space is discrete with the four-movement actions (i.e. up, right, down, left). In the following, we now show an example of the observation space of a 5x5 variation of the mazes we use throughout the paper following the OpenAI Gym interface (Brockman et al., 2016).

```
obs_shape = env.observation_space.shape

# obs_shape == (3,5,5)

obs, reward, done, info = env.step( ... )

# obs = [
  [ # avatar in these locations
    [0,0,0,0,0],
    [0,1,0,0,0],
    [0,0,0,0,0],
    [0,0,0,0,0],
    [0,0,0,0,0]
  ],
  [ # wall in these locations
    [1,1,1,1,1],
    [1,0,0,0,1],
    [1,0,0,0,1],
    [1,0,0,0,1],
    [1,1,1,1,1]
  ],
  [ # goal in these locations
    [0,0,0,0,0],
    [0,0,0,0,0],
    [0,0,0,0,0],
    [0,0,0,1,0],
    [0,0,0,0,0]
  ]
]
```

### A.6.2 DEEPSEA

The DeepSea environment is taken from (Osband et al., 2019) and has been used to evaluate the performance of intrinsic exploration objectives (Schäfer et al., 2021). DeepSea represents a hard-exploration task in a $N \times N$ grid where the agent starts in the top left and has to reach a goal in the bottom right location. At each timestep, the agent moves one row down and can choose whether to descend to the left or right. The agent observes the 2D one-hot encoding of the grid and receives a small negative reward of $\frac{-0.01}{N}$ for moving right and 0 rewards for moving left. Additionally, the agent receives a reward of +1 for reaching the goal and the episode ends after $N$ timesteps. Hence, it is very hard for an agent trained on extrinsic rewards only to solve the credit assignment problem and realize that going right is the optimal action.

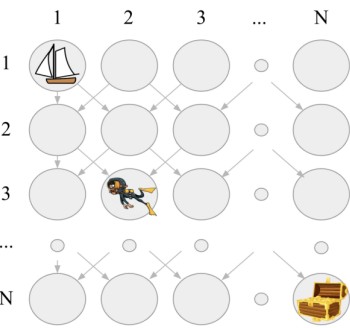

Figure 11: The DeepSea environment.

The episodes last exactly $N$ steps, and the complexity of the task can be incremented by increasing $N$.

### A.6.3 GODOT MAP

We use the Godot game engine to design the 3D world used in Section 5.1, which we open-source together with the code. We show a global view of the map in Figure A.6.3. The map is of size 120x120 and has continuous state and action spaces. To apply count-based methods we discretize the map in bins of size 5, resulting in a 24x24 object $\mathcal{N}_t$. The observations fed to the agent are the result of shooting several raycasts from the agent's perspective. The observations also contain global features like the agent's current velocity, rotation, and position. The action space contains 3 continuous dimensions that control the velocity, rotation, and jumping actions.

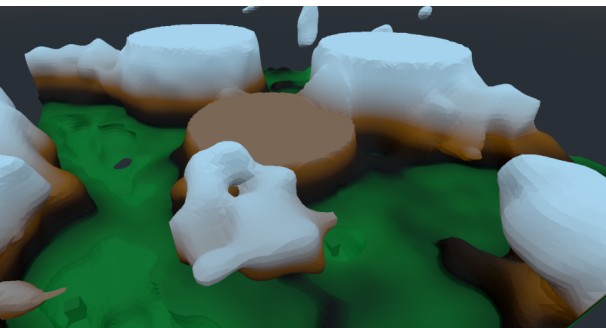

Figure 12: Global view of the 3D Godot map.

### A.6.4 MINIHACK MULTIROOM

We use the Multiroom-N6 task from the Minihack suite (Samvelyan et al., 2021) to evaluate the performance of E3B and our proposed augmentation, as originally used in Henaff et al. (2022). The environment provides pixel and natural language observations and generates procedurally generated maps at each episode. The rewards are only non-zero when the agent finds the goal location in a maze that contains 6 rooms. We use the same policy architecture described in Section C.1.1 in Henaff et al. (2022).

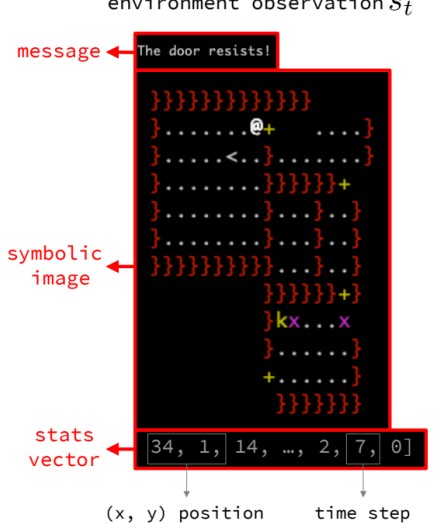

Figure 13: An observation from MiniHack.

### A.6.5 PROCGEN MAZE

We use the Procgen-Maze task from the Procgen benchmark (Cobbe et al., 2020) to evaluate the performance of E3B and our proposed augmentation. We use the *memory* distribution of mazes. The mazes are procedurally generated at each episode, have different sizes, and use different visual assets. Procgen-Maze provides pixel observations and the rewards are only non-zero if the agent finds the goal location.

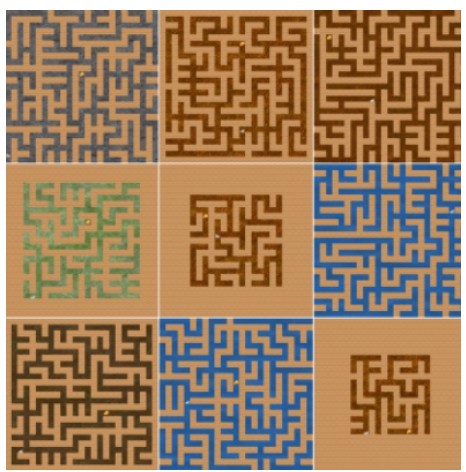

Figure 14: Procedurally generated mazes from the Procgen-Maze environment.

### A.7 STATE-ENTROPY MAXIMIZATION

In this section, we provide the pseudo-code for the surprise-maximization algorithm presented in Section 3.1.3. Note that the update rule for the sufficient statistics of the generative model $p_{\phi_t}$ is Markovian as it is updated at each timestep with the new information from the next state. As mentioned in the paper, we use a Gaussian distribution to model $p_{\phi_t}$, and hence when using SOFE, we pass its mean and standard deviation to the RL agents.

---

**Algorithm 1** Surprise Maximization

---

1: **while** not converged **do**
2:   $\beta \leftarrow \{\}$                                              ▷ Initialize replay buffer
3:   **for** $episode = 0, \ldots, M$ **do**
4:     $s_0 \leftarrow p(s_0); \tau_0 \leftarrow \{s_0\}$
5:     **for** $t = 0, \ldots, T$ **do**
6:       $a_t \sim \pi_\theta(a_t|s_t)$                                       ▷ Get action
7:       $s_{t+1} \sim T(s_{t+1}|s_t, a_t)$                              ▷ Step dynamics
8:       $r_t \leftarrow -\log p_{\phi_t}(s_{t+1})$                      ▷ Compute intrinsic reward
9:       $\tau_{t+1} \leftarrow \tau_t \cup \{s_{t+1}\}$                   ▷ Update trajectory
10:      $\phi_{t+1} \leftarrow \mathcal{U}(\tau_{t+1})$               ▷ Update mean and variance
11:      $\beta \leftarrow \beta \cup \{(s_t, a_t, r_t, s_{t+1})\}$
12:     **end for**
13:   **end for**
14:   $\theta \leftarrow RL(\theta, \beta)$                                        ▷ Update policy
15: **end while**

---

### A.8 EXPLORATION FOR SPARSE REWARDS

In this section, we show the complete set of results for Section 5.2. The results include confidence intervals and learning curves for DQN, A2C, PPO, and PPO-LSTM for the task of reaching the goal in Maze 2 in Figure 2. We also include the *partially-observable* setting, where the agent does not observe the full maze but 5x5 agent-centred observation.

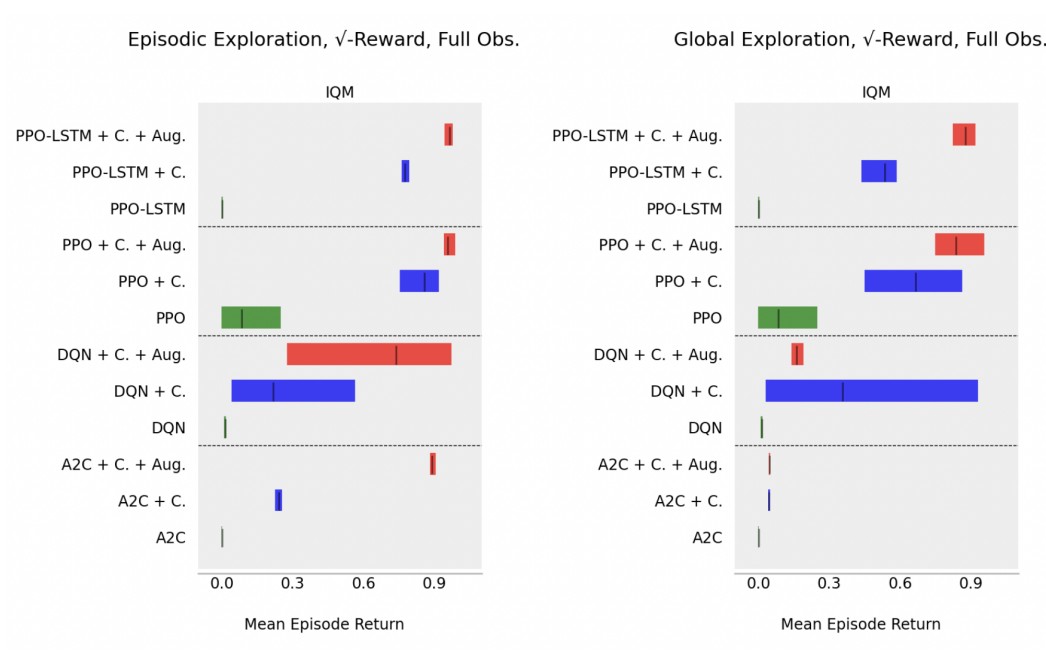

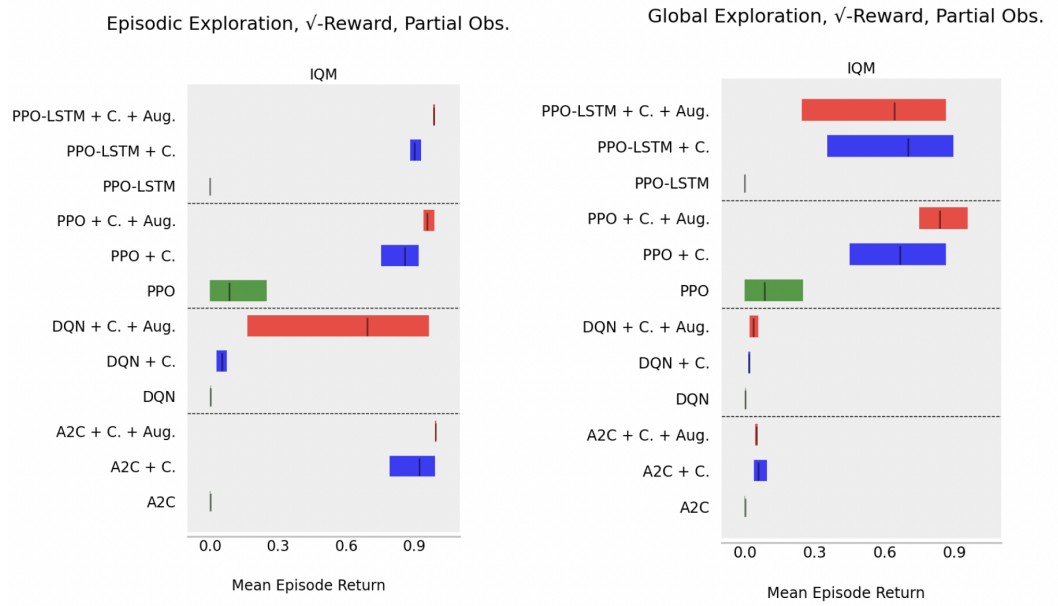

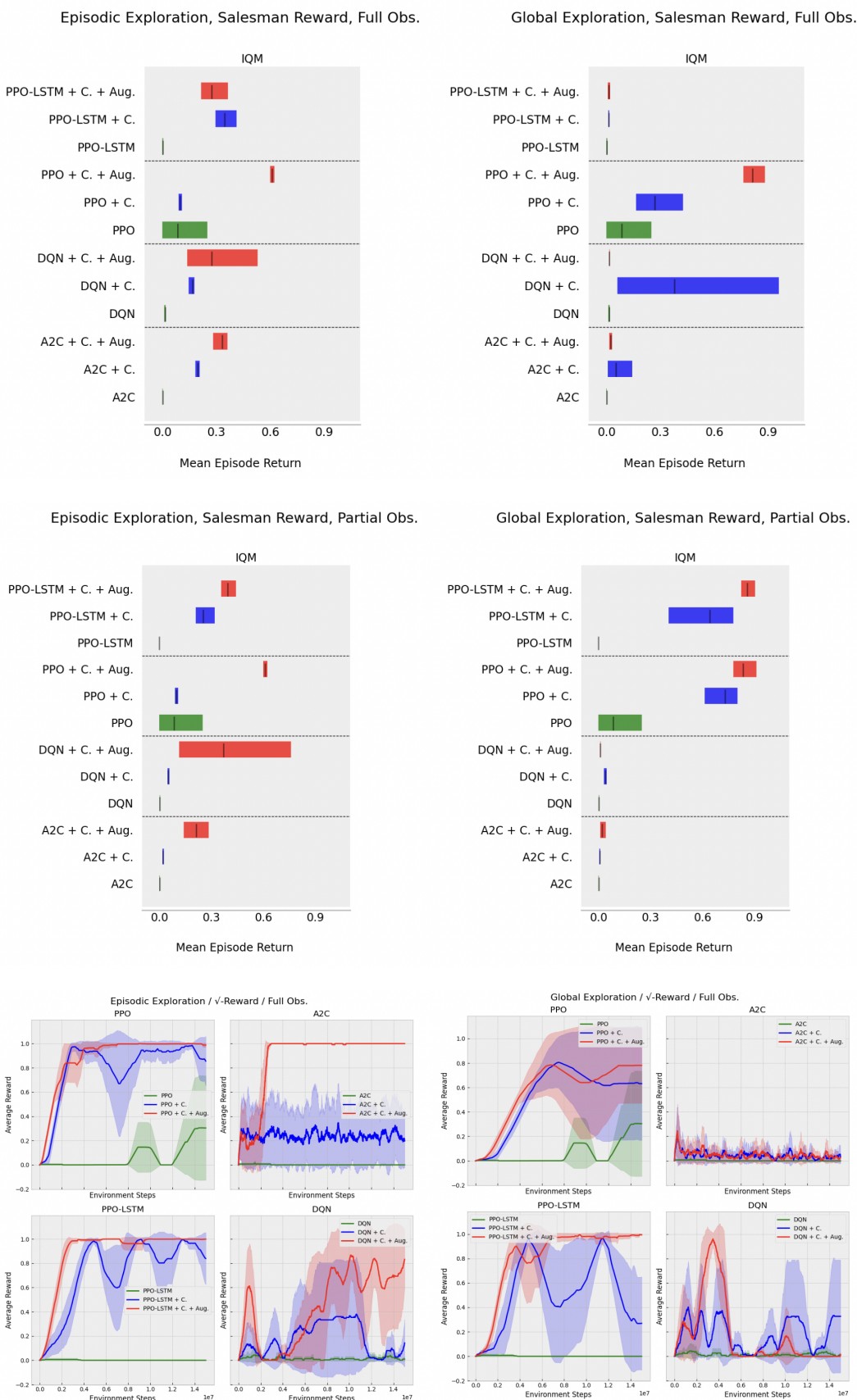

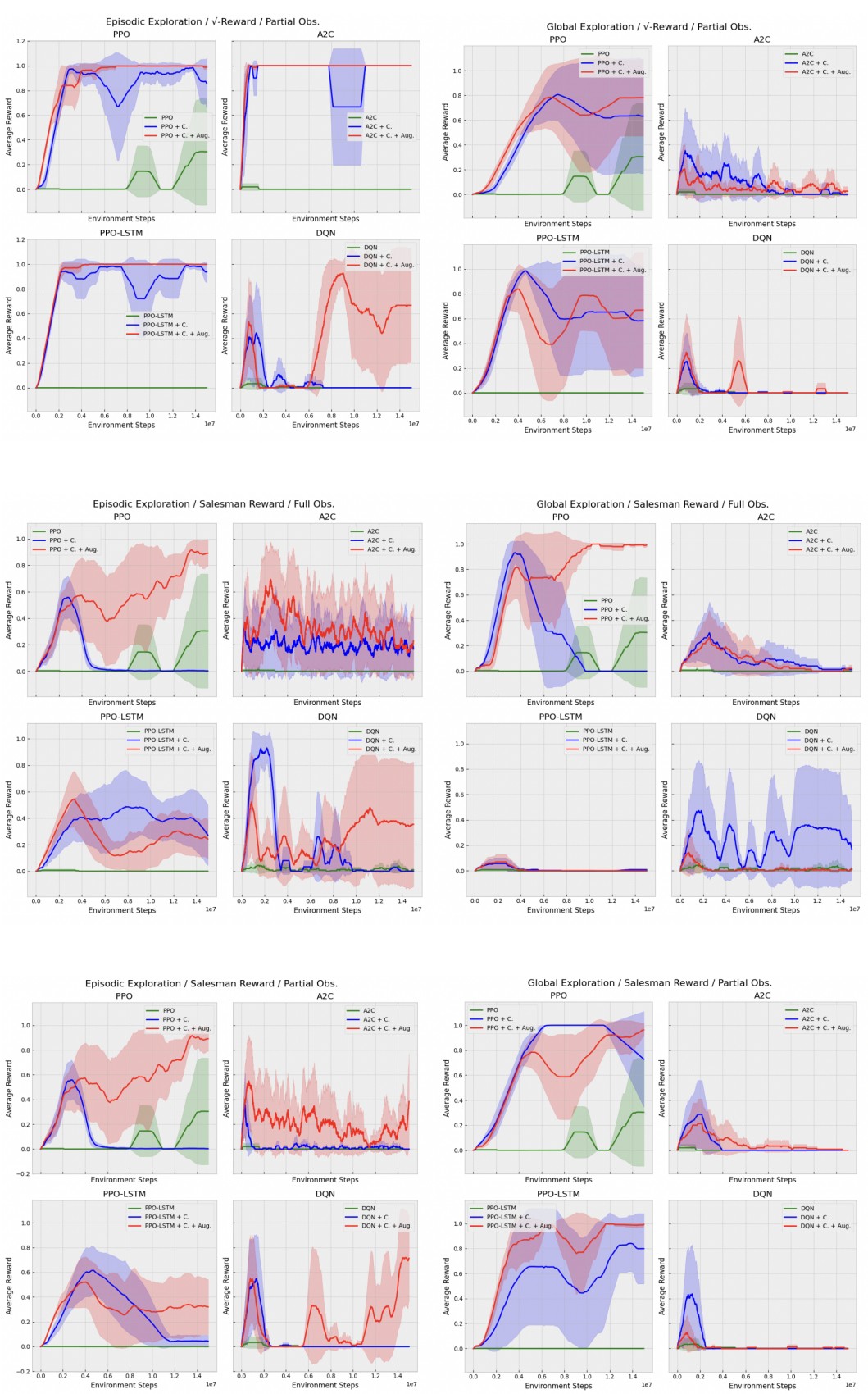

## A.9 REWARD-FREE EXPLORATION

In this section, we show the complete set of results for Section 5.1. The results include learning curves for DQN, A2C, PPO and PPO-LSTM measuring the map coverage achieved by these algorithms in the 3 mazes in Figure 2.

### A.9.1 DQN

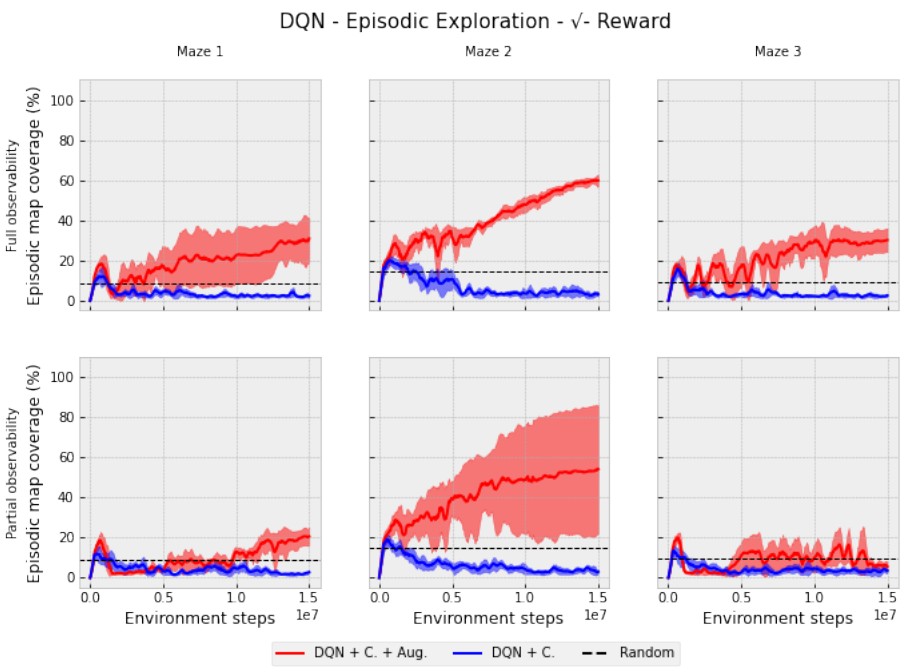

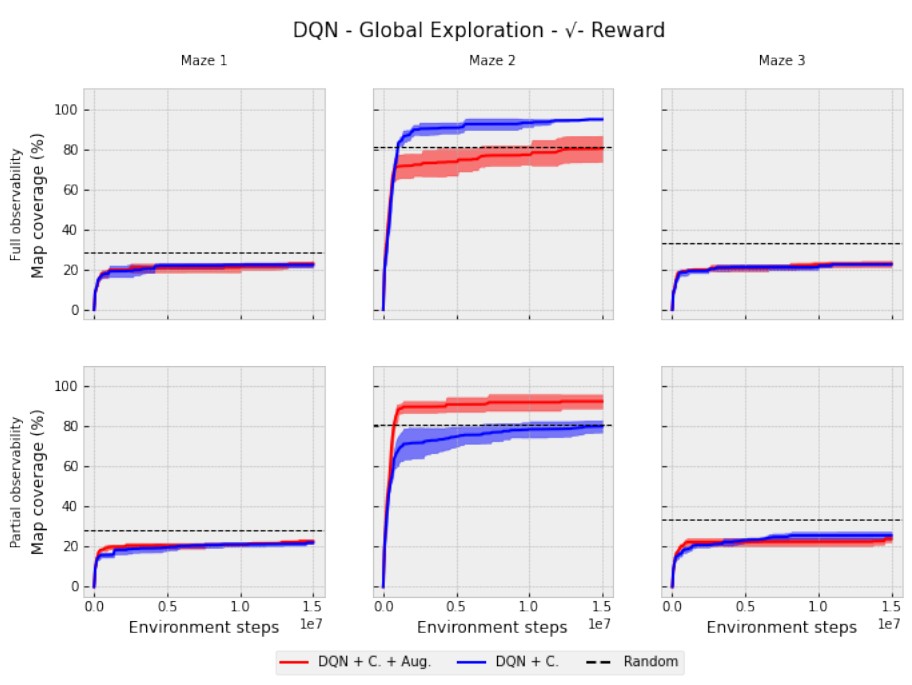

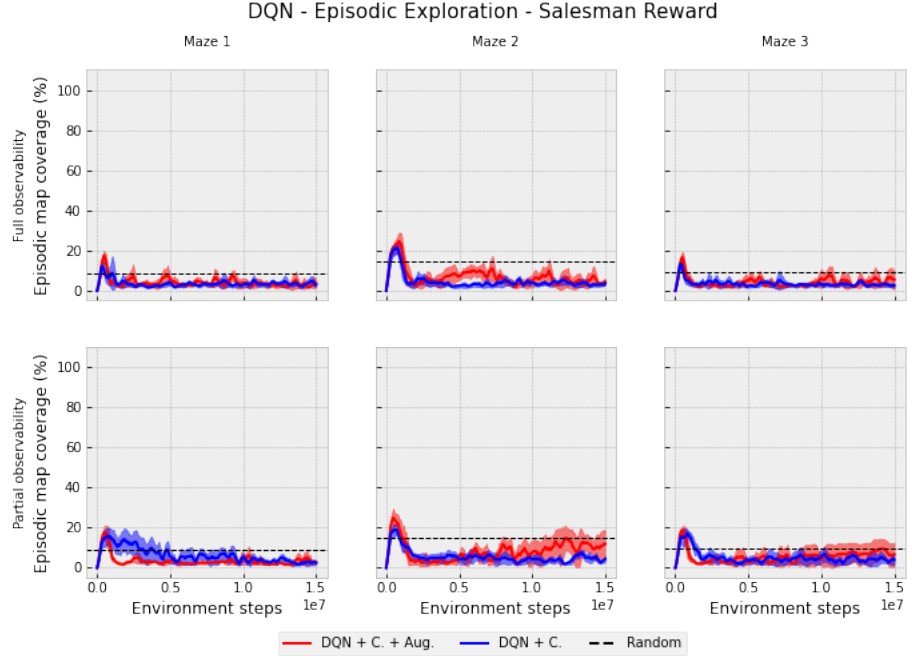

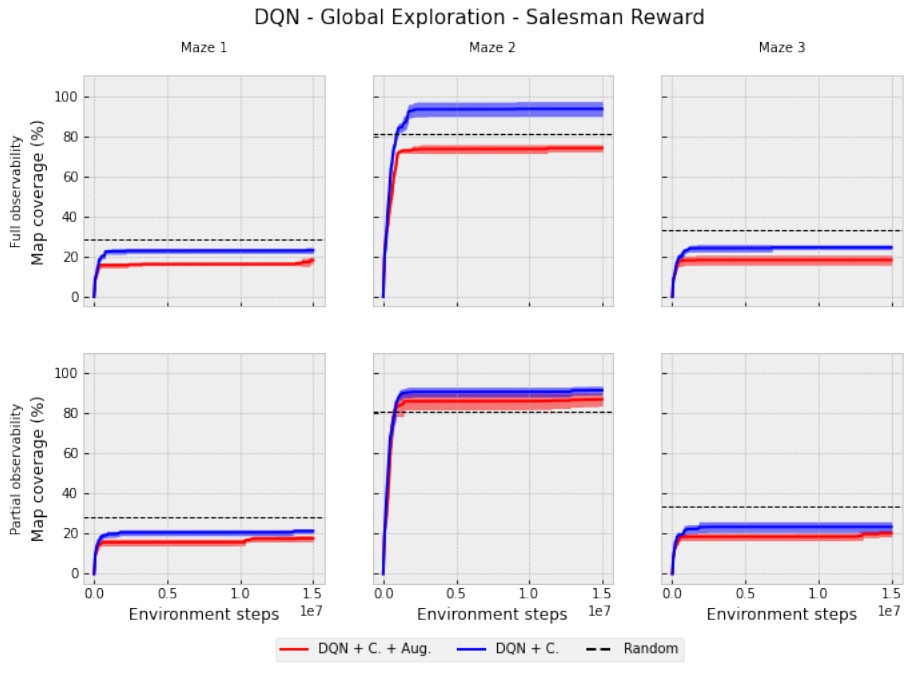

### A.9.2 A2C

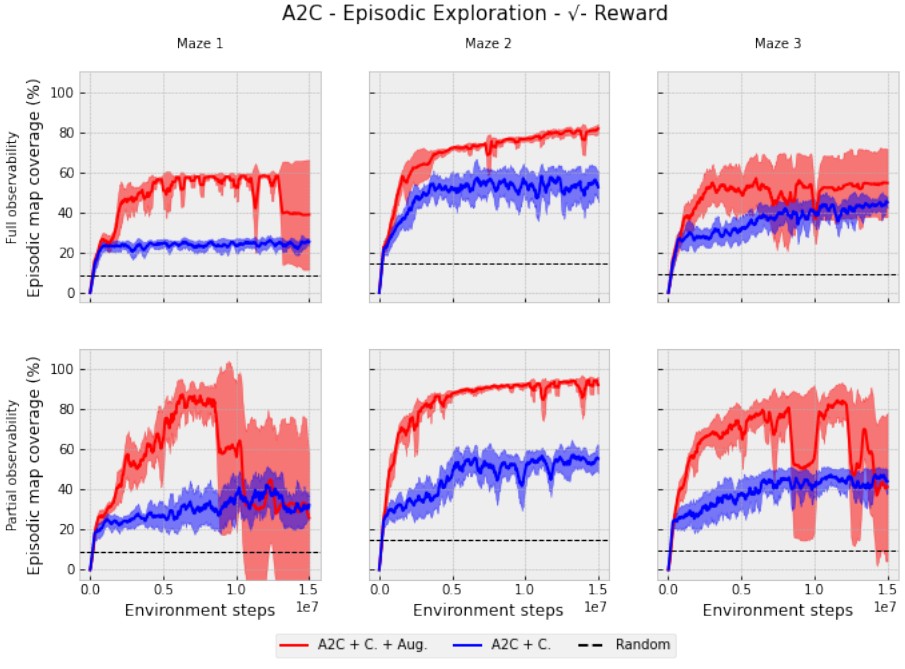

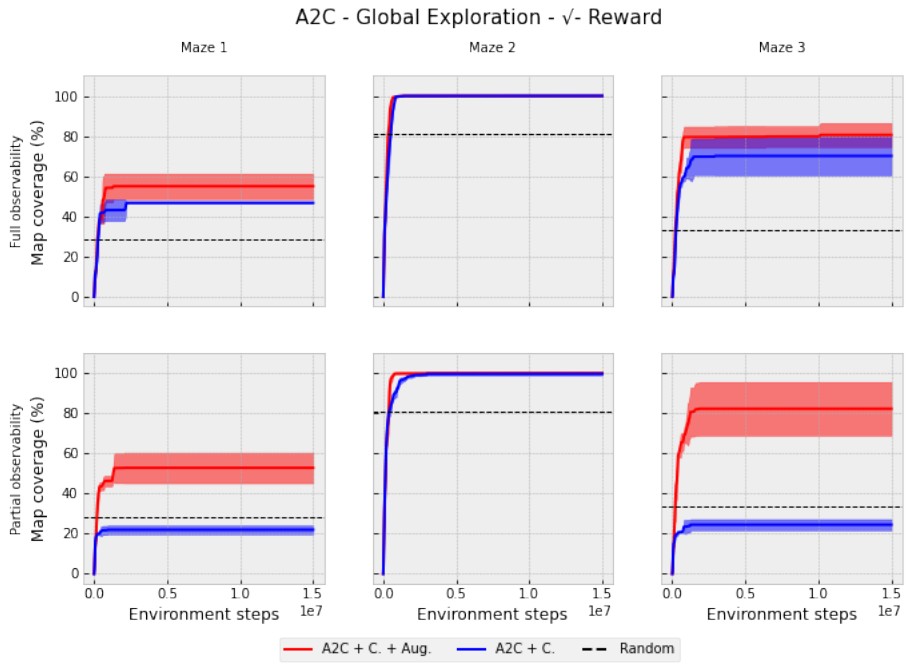

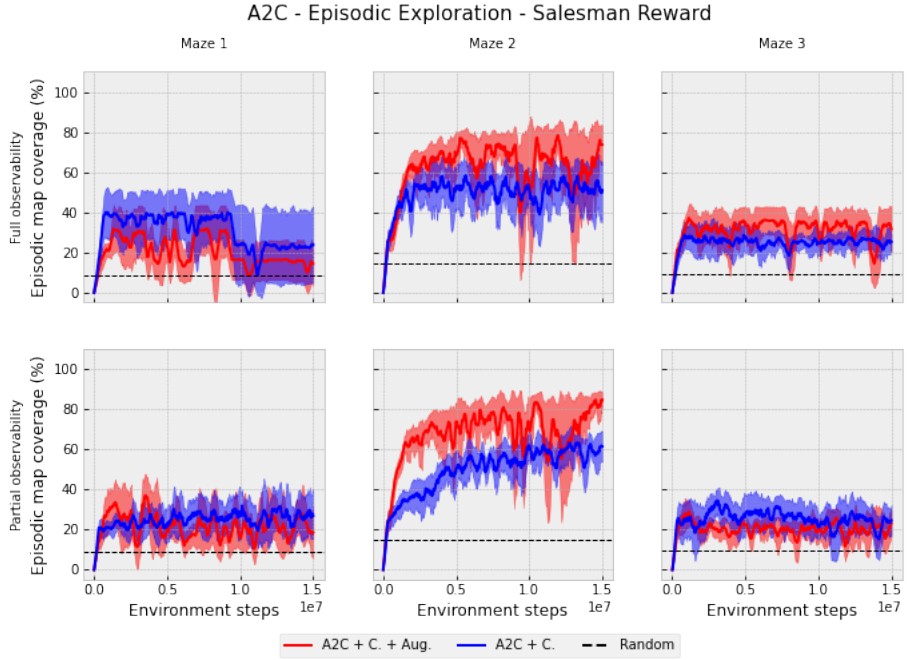

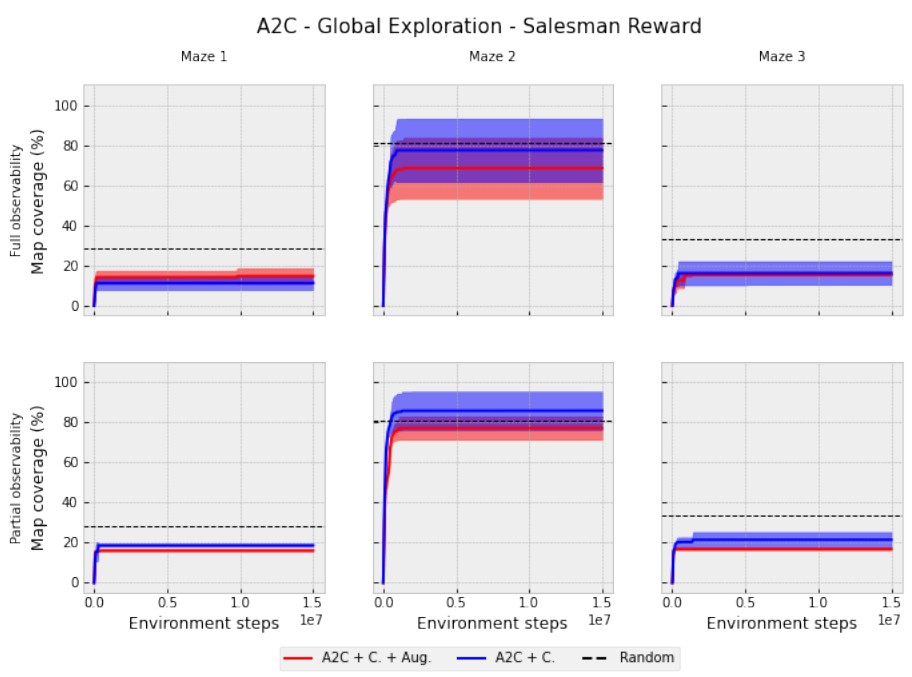

### A.9.3 PPO

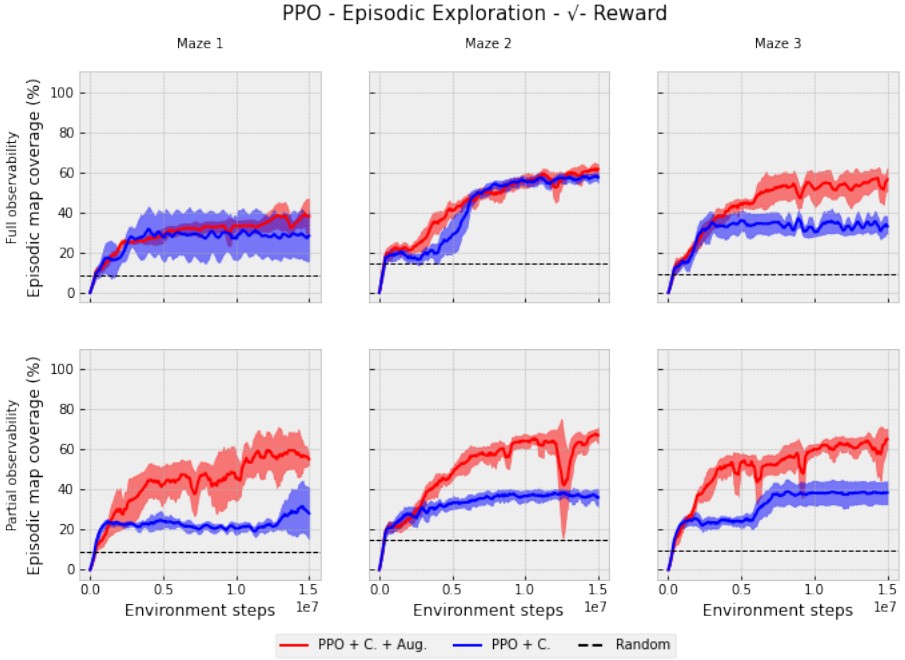

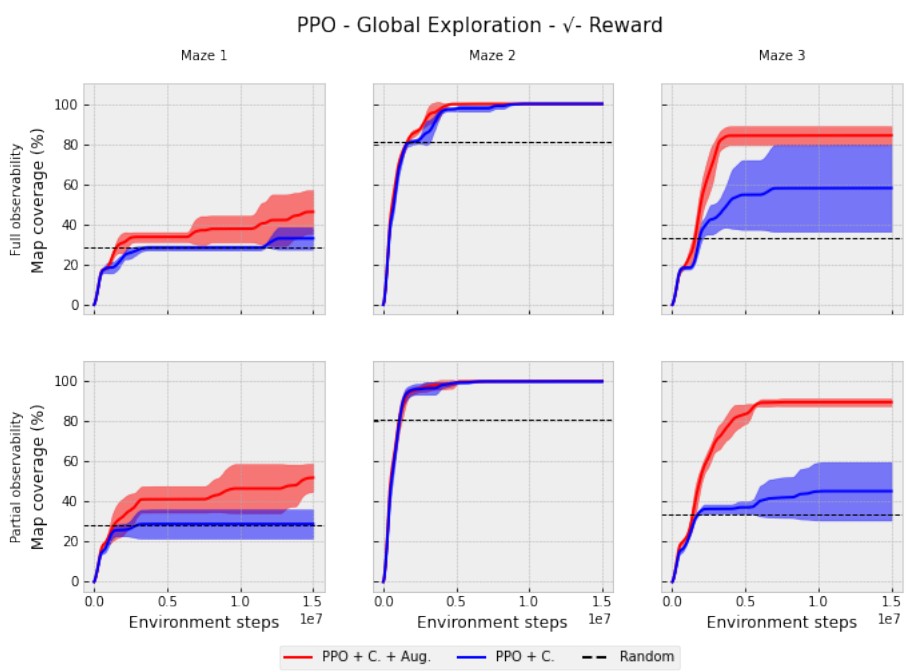

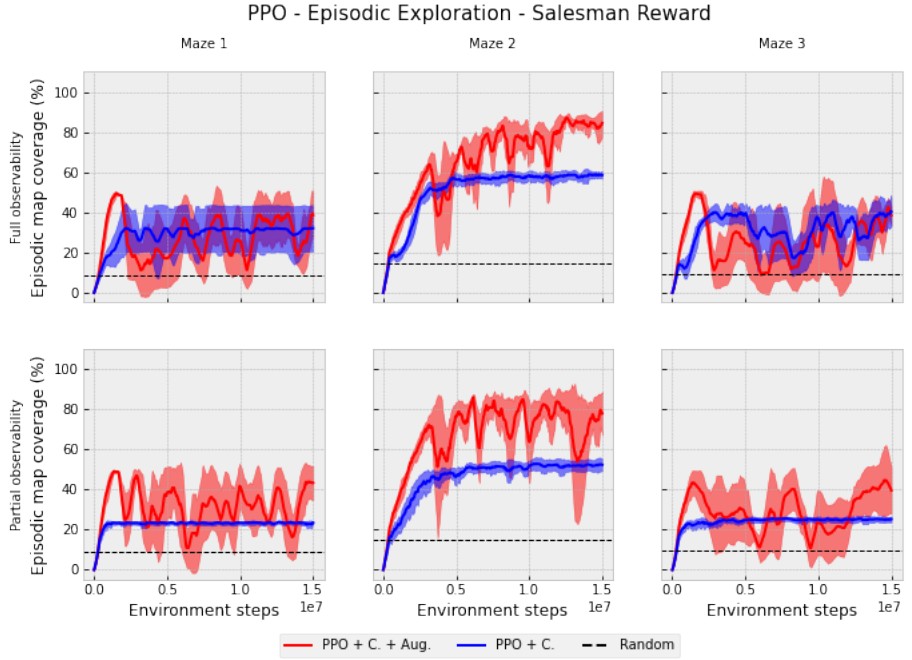

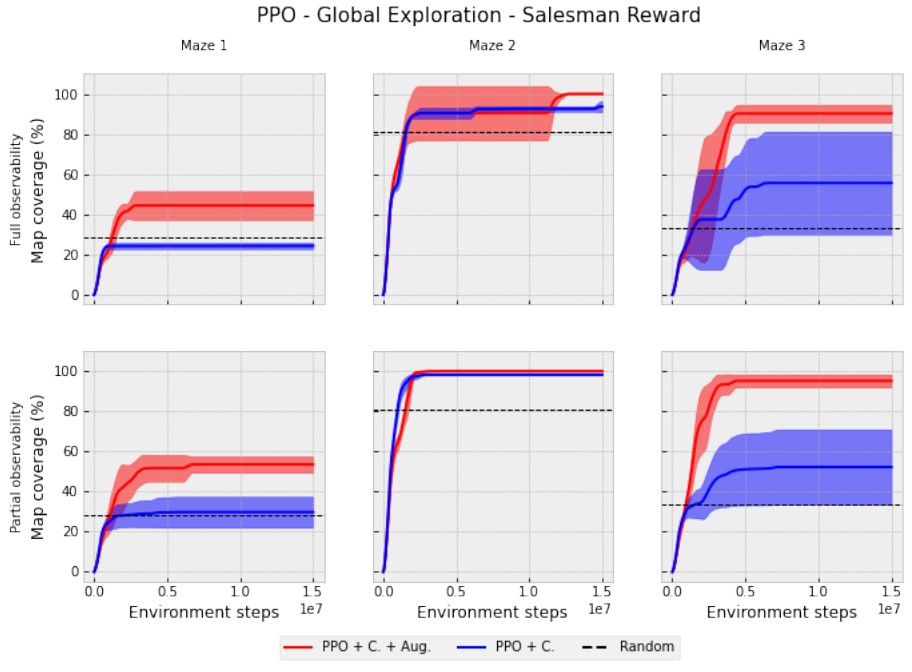

### A.9.4 PPO-LSTM

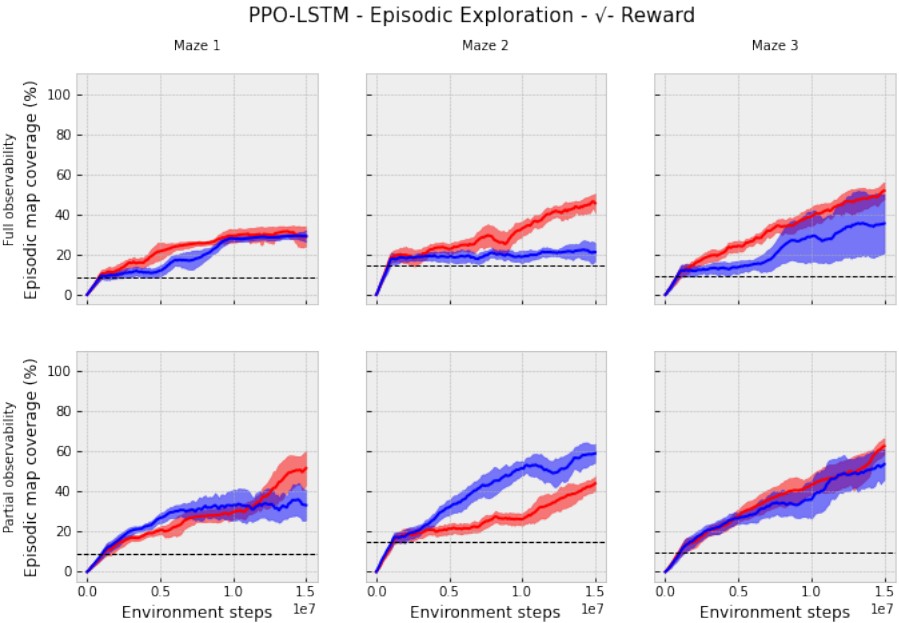

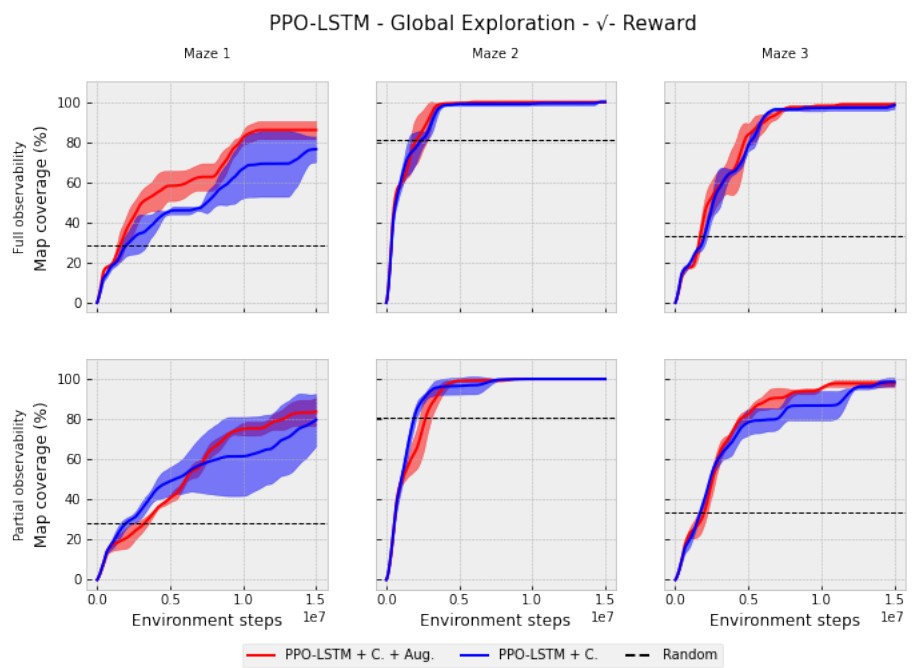

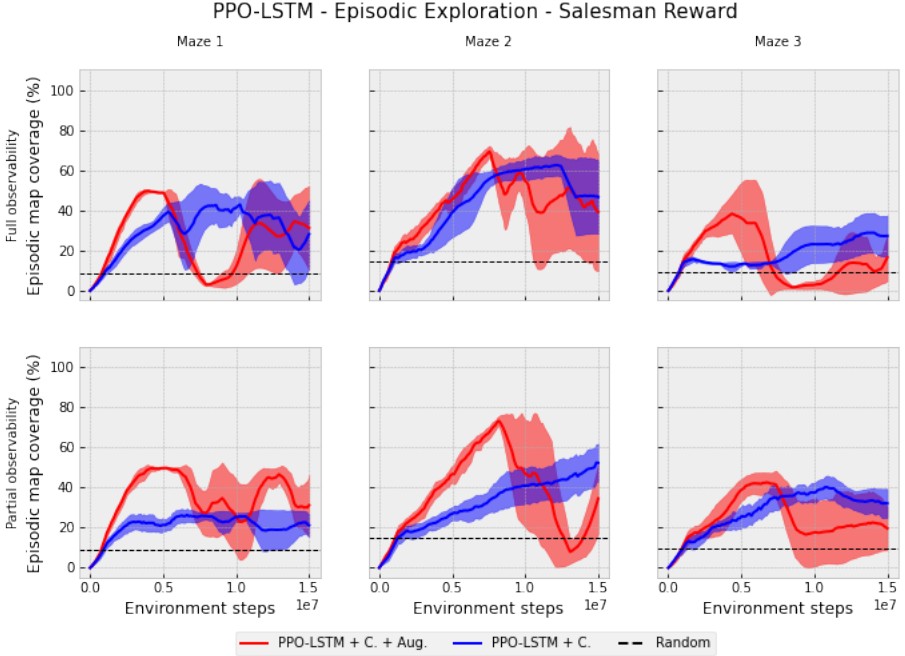

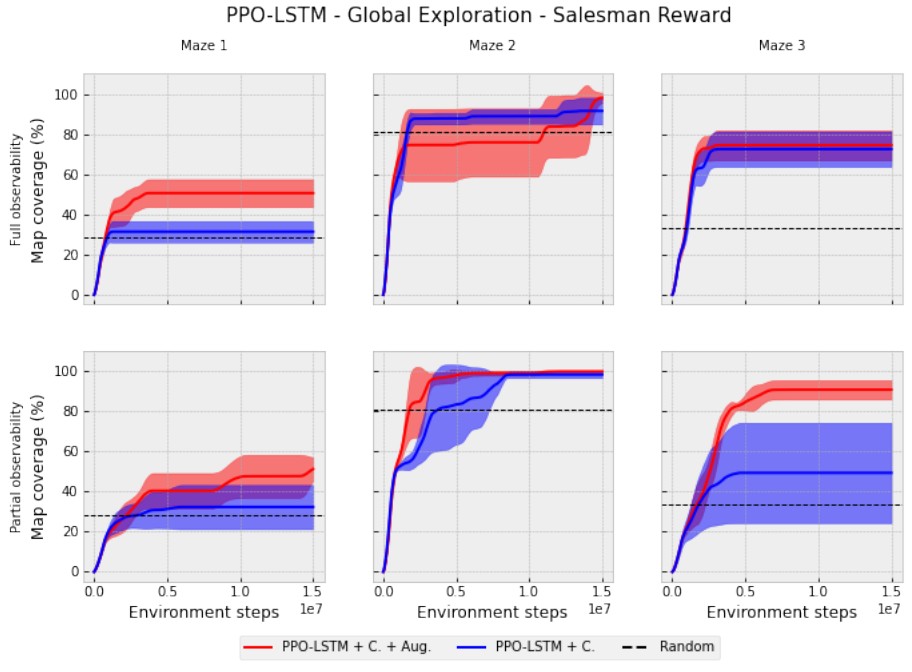

