# OpenReview forum: "Improving Intrinsic Exploration by Creating Stationary Objectives"
_ICLR.cc/2024/Conference — ICLR 2024 poster_

### Official Review · Reviewer_GnJy · 2023-10-24

**Soundness:** 3 good
**Presentation:** 2 fair
**Contribution:** 2 fair
**Rating:** 5
**Confidence:** 4

**Summary:**

This paper focuses on count-based exploration (or its variants) in reinforcement learning problems. It investigates the issue that count-based bonuses make the reward function non-stationary. The authors propose to augment the state space with visitation count for states (or representative embeddings of states). Therefore, the reward function in count-based exploration becomes stationary and satisfies Markovian property (i.e. the reward is fully determined by the state in this and next step given the augmented state space).

The authors conduct extensive experiments on 3D navigation maps, and procedurally generated environments with high-dimensional observation space. The experiments combine the proposed approach with multiple count-based exploration algorithms, on both reward-free and sparse-reward RL problems. The empirical results validate that it is beneficial for exploration to include state visitation count as a component in state space.

**Strengths:**

The proposed method is straightforward and relatively easy to implement.
The proposed method is flexible enough to be combined with many existing count-based exploration approaches.
The method is simple and effective in improving the existing count-based exploration approaches, given extensive experimental results.

**Weaknesses:**

The technical contribution is not significant enough. It is straightforward to add state visitation count as state input. When it comes to high-dimensional observation space, the proposed method is fully built on E3B and relies on its encoding of the distribution of observed embeddings. So the technical contribution is incremental upon E3B. Is E3B the best approach to solve any hard-exploration problems with high-dimensional observation space? If not, could the proposed method be general enough to improve other exploration approaches for hard-exploration problems with high-dimensional observations?

The proposed method is specifically constrained to the count-based exploration approach. As for other exploration methods with intrinsic motivations, such as RND (random network distillation), the proposed method is not compatible with RND and cannot be used to improve the exploration result.

**Questions:**

It has been mentioned several times that "we identify the matrix C_{t-1} as the sufficient statistics", but it is still vague to me. Where did you identify this? Could you provide any rigorous mathematical proof showing that C_{t-1} is a sufficient statistics for the count-based rewards?

In Section 3.2., the paper introduces the notation \phi_t without definition. For the context, I think it refers to sufficient statistics for count-based bonuses, but this point is very unclear in Section 3.2. Also, sufficient statistics in the problem setting are not defined.

In Equation 7, does s_t include sufficient statistics \phi_t or not? I think the state s_t means a fully observable state in the augmentation MDP \hat{M}, so s_t should already contain the information in \phi_t. Then it is weird to have the reward function and policy depending on both s_t and \phi_t in Equation 7.

The empirical results in Figure 4 look interesting. Could it successfully go to any goal position in the training map? If the exploration policy is trained on many different navigation maps. Could the policy be general enough to navigate to any goal position in unseen map?

---

> ### Author Response · Authors · 2023-11-14
> **Response to Reviewer GnJy**
>
> We thank the reviewer for the insightful comments. We have addressed your points and improved the paper. In line with your comments, we now better support our claims that SOFE provides performance gains across several intrinsic objectives, and have shown that it outperforms prior work like DeRL, which also attempts to stabilize training from intrinsic rewards. We now address your points:
>
> **1) Technical contribution**
>
> Regarding the contribution of the paper, we have taken steps to better express the research value of our proposed method, SOFE. We have now added experiments that show that SOFE outperforms recent algorithms like DeRL [1], which are more complex and still aim to tackle the same objective of stabilizing the training with intrinsic rewards. SOFE allows for the training of a single policy in an end-to-end fashion, introducing minimal additional complexity to the RL loop. In contrast, DeRL trains decoupled exploration and exploitation policies to attempt to stabilize the joint objective, which has several limitations: the exploitation policy must be trained from off-policy data only, the exploration policy still faces a complex optimization problem as it optimizes for non-stationary intrinsic rewards, and two completely different policies must be trained in the same loop. With this, we highlight that the simplicity of SOFE is a feature of our contribution.
>
> Please see lines 40-48, 103-106, 256-263, 310-314, and Table 1.
>
> It is true that a version of SOFE uses E3B as a way to estimate the true state distribution. In fact, this is part of the main contribution of SOFE, finding and using the best distributions to estimate the visitation frequencies that are easy for deep models to understand. In larger continuous spaces we use E3B to help estimate the state distribution. Furthermore, E3B achieved SOTA results in partially observable, high-dimensional environments like MiniHack and Habitat (**which we have improved with SOFE**), further outperforming popular baselines like RND and ICM. To the best of our knowledge, there is no published article that reports better exploration performance than E3B in such challenging environments.
>
> Please see the improved Section 3.2.2. Please see Section 4.1 - Intuition here
>
> Regarding RND, we are now working to add a discussion section that motivates future work that extends SOFE to other exploration bonuses like RND and ICM, as we had discussed this before and believe it is possible.
>
> **2) The proposed method is specifically constrained to the count-based exploration approach**
>
> We have disentangled SOFE from count-based bonuses, and have shown that SOFE provides orthogonal gains across exploration objectives of different nature like state-entropy maximization and pseudo-counts.
>
> Please see lines 13-16, 54-58, 92-93, and the improved Section 3.2.
>
> **Q1) Identifying the sufficient statistics**
>
> We thank the reviewer for raising this point. We have modified the paper to better express that SOFE augments the observations with all the moving components of the intrinsic rewards, and hence solves their non-stationarity. Concretely, the state-visitation frequencies are the only dynamically changing components in Equations 1 and 2, and so are the ellipsoid matrix in Equation 3, and the parameters of the generative model in Equation 5. We have modified the paper to better explain this point.
>
> Please see the improved Section 3.2.
>
> An example where it is easy to see that the state-visitation frequencies are sufficient statistics that allow for the existence of an optimal Markovian stationary policy is the following: Consider an MDP with paths S1 → S2 → S3 and S3 → S2 → S1 (no direct path between S3 and S1). The agent always starts at S1. To optimize for the count-based reward $R(s) = \frac{1}{N(s)}$ the optimal policy is to go S1→S2→S3→S2→S1→S2→S3 – but of course, this is not a stationary policy (needs to swap actions in S2 depending on the counts of S1 and S3). The best stationary policy here is S1→S2, S3→S2 and then a 50/50 chance from S2→S1 and S2→S3 – but this is suboptimal. The optimal policy described above can only be learned with access to the state-visitation frequencies (provided by SOFE):  if the agent observed the counts then the optimal policy is stationary: at S2, if N(S3) > N(S1) go to S1, else go to S3.
>
> **Q2) Presentation & notation**
>
> We thank the reviewer for spotting this important issue. We have modified Section 3.2 to improve the notation and better present SOFE.
>
> Please see the improved Section 3.2.
>
> **Q3) Augmented states and $\phi_t$**
>
> We thank the reviewer for raising this point. We have removed the equation for the optimal policy in the augmented MDP, as it did not provide valuable insights about SOFE. However, we have kept the definition of the augmented MDP where, as you correctly pointed out, the states $\hat{s}_t \in \mathcal{\hat{S}}$ contain both the observations and $\phi_t$.
>
> In the next question, we answer the remaining questions.

---

> ### Author Response · Authors · 2023-11-17
>
> **Q4) Analysis of the behaviours learned by SOFE**
> We thank the reviewer for their comments on these experiments. While we believe that the new experiments that you suggest are relevant, we have moved these experiments to the Appendix section in an effort to make space in the main paper for more important comparisons to related work as suggested by other reviewers. However, we have framed these experiments as an in-depth analysis of the behaviours learned by SOFE, which provide valuable insights into how SOFE enables the agents to use the augmented information to drive efficient exploration.
>
> **Q4.1) Could the policy be general enough to navigate to any goal position in an unseen map?**
>
> Given the interpolation properties of deep networks, we hypothesize that if SOFE is trained over enough combinations of maps, then such generalization would emerge. This should occur in a similar fashion to goal-conditioned RL methods or other contextual MDPs. Importantly, note that in Procgen-Maze and Minihack, every episode reset creates a new map (previously unseen during training since it is procedurally generated). In this case, our results show that SOFE is **necessary** for agents to solve the sparse-reward tasks (see Figure 7) which demonstrates that SOFE allows agents to make smart use of the augmented information in the states.
>
> We again thank the reviewer for the valuable feedback and we believe we have addressed all concerns from the reviewer.

---

> > ### Author Response · Authors · 2023-11-20
> >
> > Dear Reviewer,
> >
> > We hope that you've had a chance to read our responses and clarifications. As the end of the discussion period is approaching, we would greatly appreciate it if you could confirm that our updates have addressed your concerns.
> >
> > Thank you very much.

---

> ### Comment · Reviewer_GnJy · 2023-11-20
> **Thanks for the detailed response**
>
> Thank authors for the detailed response. I appreciate your efforts and see improvements in the paper, especially Section 3.2. But two of my concerns are not fully addressed. 1) combination of the proposed method of RND/ICM (it will be more convincing if there is experimental results supporting the benefits of proposed method). 2) the generalization ability of the learned policy for more goal positions (it will be more impressive if the learned policy can reach any goal on unseen maps). So I'd like to keep my score.

---

> > ### Author Response · Authors · 2023-11-21
> >
> > **Q1) “combination of the proposed method of RND/ICM (it will be more convincing if there is experimental results supporting the benefits of proposed method)”**
> >
> > Importantly, SOFE is designed to overcome the non-stationarity of intrinsic rewards by finding a simple representation of the state visitation-frequencies or their representative embedding. RND and ICM **do not** estimate the state-visitation frequencies, as these methods use prediction-error as a heuristic. It is possible to combine SOFE with RND or ICM. However, to model RND or ICM, SOFE needs the parameters of the distribution that is computing the intrinsic rewards. In the case of RND and ICM that distribution is defined by the complete set of parameters of the neural networks being trained (i.e. the predictor network in RND and the forward/inverse dynamics models in ICM). It is very difficult to describe the sufficient statistics for such high-dimensional and non-linear distributions, and this makes the non-stationarity of the RND and ICM rewards practically unsolvable. In our work, we put focus on more principled exploration methods that aim to use the state-visitation frequencies (e.g. counts, state-entropy maximization) or estimate them (e.g. E3B). When using these methods, SOFE strikes a middle ground where the policy can be conditioned on a representation that is used to compute the state-visitation frequencies and leverage it to drive exploration more efficiently as we show in our extensive experiments.
> >
> > Additionally, recent works on pseudo-count-based methods have achieved great results in popular benchmark environments by leveraging the estimates of the state-visitation frequencies, and RND and ICM fall short in many challenging environments (e.g. stochastic or procedurally generated). [1, 2, 3]
> >
> >
> > [1] Henaff, Mikael, et al. "Exploration via elliptical episodic bonuses." Advances in Neural Information Processing Systems 35 (2022): 37631-37646.
> > [2] Lobel, Sam, Akhil Bagaria, and George Konidaris. "Flipping Coins to Estimate Pseudocounts for Exploration in Reinforcement Learning." arXiv preprint arXiv:2306.03186 (2023).
> > [3] Wang, Kaixin, et al. "Revisiting intrinsic reward for exploration in procedurally generated environments." The Eleventh International Conference on Learning Representations. 2022.
> >
> >
> > **Q2) "the generalization ability of the learned policy for more goal positions (it will be more impressive if the learned policy can reach any goal on unseen maps)"**
> >
> > We believe that the requested experiments are not aligned with the contributions of our work. In these experiments, which are in the Appendix, we provide insightful details about how agents leverage SOFE to drive exploration efficiently. We show that agents reach multiple goals without having been trained in a goal-conditioned manner. This indicates that thanks to SOFE, agents can make efficient use of the augmented information to drive exploration efficiently toward the unvisited states.
> >
> > The capability of generalizing across multiple maps and goals is related to the literature of goal-conditioned RL, and prior work has shown that when trained on a large enough goal space, agents generalize to previously unseen goals. However, we note that this analysis is a use case of SOFE, but not the focus of our work, and is unrelated to SOFE's ability to stabilize and improve intrinsic objective training.
> >
> > We thank the reviewer for the feedback and we have addressed all concerns from the reviewer.

---

> > > ### Author Response · Authors · 2023-11-22
> > >
> > > Dear Reviewer,
> > >
> > > We hope that you've had a chance to read our responses to your remaining concerns. As the end of the discussion period is in less than 24 hours, we would greatly appreciate it if you could confirm that our updates have addressed your concerns, and we invite you to re-evaluate the paper if that is the case.
> > >
> > > Thank you very much.

---

### Official Review · Reviewer_Yao3 · 2023-10-30

**Soundness:** 2 fair
**Presentation:** 3 good
**Contribution:** 2 fair
**Rating:** 6
**Confidence:** 3

**Summary:**

This paper studies intrinsic motivation for RL. First, the authors notice that common intrinsic bonuses (count-based is considered as the illustrative instance) induce non-stationary objectives, which can make the learning process unstable. Then, they propose a solution to this problem by designing a method, called SOFE, to make the objective stationary through state augmentation. Finally, they empirically evaluate the proposed algorithm in a variety of domains, including 3D navigation and ProcGen tasks.

**Strengths:**

- (Originality) Although limitations of count-based bonuses for intrinsic exploration have been considered before, the algorithmic idea of augmenting the state representation to make the objective stationary is new to the best of my knowledge;
- (Significance) The experimental results look promising at least and experiments are carried out in some challenging and interesting domains, such as ProcGen Maze;
- (Clarity) The main ideas of the paper are presented with clarity, but most of the relevant implementation details are deferred to a very brief section in the supplementary (Appendix A.5).

**Weaknesses:**

- (Motivation) The paper does not clarify that most of the reported considerations are valid when count-based bonuses are the actual learning objective rather than shaping of (sparse) external rewards. The former is arguably not the setting count-based methods have been designed for;
- (Novelty and scope) Identifying the non-stationarity of the bonuses as a limitation of count-based methods does not look to be novel (e.g., Schafer et al., Decoupled reinforcement learning to stabilise intrinsically-motivated exploration, 2022) and the paper does not credit previous works on that;
- (Robustness of the empirical evaluation) The experiments section does not specify how count-based bonuses are implemented. Several strategies exist, and they are known to make a significant difference in the resulting performance, which leaves one wondering how general the reported comparison really is. Moreover, the paper does not compare SOFE against alternative pure exploration methods aside from E3B, ICM, RND in Fig.7.

**EVALUATION**

While the paper is overall interesting, as it tackles a relevant problem on the inherent non-stationarity of count-based bonuses and provides some promising experimental results, I think it is falling short of the quality required for an ICLR paper in terms of clarity of the motivation, discussion of prior works, and robustness of the empirical evaluation. Hence, I am currently providing a slightly negative evaluation, while I report some detailed comments and questions below.

**Questions:**

1) Count-based bonuses have been largely employed for hard-exploration tasks, either in terms of regret minimization in theoretical papers or shaping of sparse rewards in empirical literature. From my perspective, count-based bonuses are not a great fit for pure exploration instead: They are not only non-stationary, as the authors noted, but also vanishing, and suffer from some other well-known issues (see Ecoffet et al., Go-Explore: A new approach for hard-exploration problems, 2021). However, only in the pure exploration setting the POMDP argument and optimality of history-based policies seem to make sense. If an external reward is there, it is a actually good to converge to a Markovian deterministic policy. Can the authors comment on why they think it is worth studying count-based methods for pure exploration?

2) To follow-up the previous question, count-based bonuses have been used in recent reward-free RL literature as well (e.g., Jin et al., Reward-free exploration for reinforcement learning, 2020). Also in the latter setting, the bonuses are freezed at the start of the episode, so that a Markovian policy optimizing a stationary objective is always deployed to collect data.

3) To address pure exploration settings, other stationary objectives have been designed, e.g., state entropy maximization. Can the authors relates their contributions with this stream of works:
- Hazan et al., Provably efficient maximum entropy exploration, 2019;
- Mutti et al., Task-agnostic exploration via policy gradient of a non-parametric state entropy estimate, 2021;
- Liu & Abbeel, Behavior from the void: Unsupervised active pre-training, 2021;
- Seo et al., State entropy maximization with random encoders for efficient exploration, 2021;
- Yarats et al., Reinforcement learning with prototypical representations, 2021;
- and many others.
Moreover, is state entropy maximization a meaningful baseline for the reported experimental evaluation?

4) While I am not particularly familiar with the literature of count-based methods, my feeling is that this kind of bonuses have been deeply studied. From a brief research, I am not sure the authors made a thorough due diligence of prior works, especially those addressing the limitations of count-based methods (e.g., Shafer et al., 2020 and Ecoffet et al., 2021 mentioned before). Interestingly, previous works also showed that $1/n$ bonuses bring faster learning in pure exploration settings (Menard et al., Fast active learning for pure exploration in reinforcement learning, 2021).

5) How are the pseudocounts implemented in the experiments? This seems to make a huge different. For instance, Bellemare et al. (2016), Ostrovski et al., Count-based exploration with neural density models (2017), Tang et al., (2017), Machado et al., Count-based exploration with the successor representation (2020) all make different design choices with varying results. Do the authors think their results are general for every implementation of count-based bonuses or for just one?

6) How many seeds are considered in the experiments and what is the meaning of the shaded areas? This crucial information seems to be missing for some of the results.

7) In the ProcGen Maze experiment, the paper comapres SOFE with E3B, ICM, RND. These does not seem to be the state of the art for procedurally generated tasks. Do the authors considered also tailored methods, such as (Ghosh et al., Why generalization in RL is difficult, 2021; Zisselman et al., Explore to generalize in zero-shot RL, 2023)?

---

> ### Author Response · Authors · 2023-11-13
> **Response to Reviewer Yao3 - Weaknesses**
>
> We thank the reviewer for the insightful comments. We have addressed your points and improved the paper. We invite you to go over the changes in the updated version of the paper. Overall, we now better highlight the generality of SOFE, providing results across multiple intrinsic exploration objectives (counts, pseudo-counts, and state-entropy maximization). Importantly, and in line with your comments, we have adjusted the claims of our paper, and better positioned our work with respect to previous related research. We have also included relevant experiments that allow for necessary comparisons to previous methods, as you rightfully suggested. In the new version of the paper, we credit previous research that identifies the non-stationarity of count-based rewards, while claiming that SOFE provides performance gains across several intrinsic objectives, and outperforms prior work like DeRL, which also attempts to stabilize training from intrinsic rewards. We now address your spotted weaknesses:
>
> **1 - Motivation)** We appreciate the reviewer's attention to the motivation behind our work. While we present SOFE in a reward-free setting to illustrate its core concepts in Section 4, our method extends beyond this setting. In Figure 1, we explicitly present SOFE within a broader context, considering both intrinsic and extrinsic rewards. Initially, our focus on the reward-free setting addresses our first research question, demonstrating that SOFE effectively mitigates the non-stationarity inherent in intrinsic rewards. Next, in our second research question, we show that SOFE better optimizes the joint objective, which is the combination of task and intrinsic rewards. As rightly pointed out by the reviewer, intrinsic rewards are often used to shape sparse task rewards. In this work, we do not contradict this idea, and study both the reward-free and joint objective settings to showcase the empirical performance of SOFE in both settings. Moreover, we have modified the paper to highlight that SOFE is a general framework that works across exploration objectives of different natures (we have unified counts, pseudo-counts, and state-entropy maximization under the same framework).
>
> Please see lines 47-48, the improved Section 3.2, and lines 249-263
>
> Furthermore, in response to the experiments suggested by Reviewer 7vAW (see W2), we augment our evidence by demonstrating SOFE's efficiency in solving more popular sparse-reward tasks. Importantly, our results highlight the efficacy of SOFE beyond purely reward-free settings. We hope this clarification provides a comprehensive understanding of the motivation behind our approach.
>
> Please see lines 310-314, and Table 1.
>
> **2 - Novelty and Scope)** We appreciate the reviewer's comments regarding the novelty and scope of our work. In response to this feedback and in alignment with Reviewer 7vAW's suggestions (see W1 and W2), we have taken steps to better position our work in the context of related literature. Concretely, we have adjusted the claims of identifying the non-stationarity of rewards but we provide SOFE as a simple and high-performing framework to solve this issue.
>
> Please see lines 4-5, 40-48, 103-106.
>
> As the reviewer pointed out in the work of Schafer et al. [1], we have now included a quantitative comparison between DeRL and SOFE. This comparison serves to highlight the distinct advantages of SOFE in terms of simplicity and performance. Specifically, we demonstrate that SOFE achieves superior performance with significantly less complexity compared to DeRL. SOFE allows for the training of a single policy in an end-to-end fashion, introducing minimal additional complexity to the RL loop. In contrast, DeRL trains decoupled exploration and exploitation policies to attempt to stabilize the joint objective.
>
> Please see lines 310-314, and Table 1.
>
> [1] Schafer et al., Decoupled reinforcement learning to stabilise intrinsically-motivated exploration, 2022
>
> **3 - Robustness of the Empirical Evaluation)** We appreciate the reviewer's observation regarding the implementation of count-based bonuses and the selection of alternative baseline exploration methods for comparison. In response to your feedback and to provide greater clarity to the reader, we have modified Section 3 to provide details of the implementation of count-based bonuses
>
> Please see the improved Section 3.2 and the titles in Figures 4-6.
>
> In terms of alternative exploration methods, we have used RND and ICM, which are the most popular algorithms for long-horizon exploration in deep RL, serving as commonly used baselines in the literature. Additionally, we have introduced E3B and DeRL to broaden the comparisons.
>
> Finally, we have modified the Appendix section to enhance the clarity of our empirical evaluation and now provide implementation details for the networks and environments used in the paper. This includes hyperparameters, pseudocode, Figures, and curves.
>
> In the next comment, we tackle your questions.

---

> ### Author Response · Authors · 2023-11-13
> **Response to Reviewer Yao3 - Questions**
>
> **Q1) Count-based bonuses as the only learning objective**
>
> We use the reward-free setting as a strategic choice to illustrate the POMDP argument and the optimality of history-based policies. By studying the non-stationarity of intrinsic rewards and framing it as a POMDP, we emphasize that even in the presence of task rewards, the joint objective formed by the combination of intrinsic and task rewards remains non-stationary.
>
> It is crucial to note that the optimal policy for a joint objective involving both intrinsic and task rewards differs from the optimal policy when considering extrinsic rewards alone. When incorporating the intrinsic signal to shape task rewards, we effectively modify the MDP to facilitate the learning process, although with the trade-off of potentially obtaining policies that may not be optimal in the original task MDP. Importantly, the modified MDP, which integrates the joint objective of intrinsic and task rewards, remains non-stationary. Therefore, SOFE is applicable and beneficial in this context, addressing the challenges posed by non-stationarity and enabling agents to optimize efficiently for the joint objective.
>
> We have modified the paper to better express that SOFE does not only work under the assumptions of count-based rewards being the only learning objective. Concretely, we have unified counts, pseudo-counts, and state-entropy maximization under the same framework and have provided more empirical evidence that SOFE provides performance gains when training on a joint objective of intrinsic+task rewards.
>
> Please see lines 40-48, 256-263, 291-295, 310-314 and Table 1.
>
> **Q2) Freezing counts to stabilize the bonus**
>
> We acknowledge that this method can facilitate optimizing for the non-stationary count-based rewards.  However, this approach diverges from the exact count-based exploration objective. In contrast, SOFE does not require any modifications to the count-based objective. Importantly, the optimal policies obtained with SOFE remain optimal for the original count-based objectives, as argued in Section 4 of the paper.
>
> Please see lines 229-231.
>
> **Q3) SOFE & State-Entropy maximization**
>
> We thank the reviewer for raising this point. Even though the comparison between state-entropy maximization and count-based bonuses remains out of the scope of this paper (i.e. in terms of how much they facilitate state coverage and exploration for sparse reward tasks), we believe that SOFE can be easily applied to state-entropy maximization algorithms and provide orthogonal gains. We have now taken steps to show this for the Surprise-Maximizing algorithm [1] which fits a generative model over the state visitation distribution in order to compute its entropy and maximize it. We have modified several sections of the paper to better highlight the generality of SOFE, and have obtained empirical results that support our claims.
>
> Please see lines 13-16, 54-58, 71-73, 92-93, Section 3.2.3, lines 291-295, Figure 5.
>
> [1] Berseth, Glen, et al. "Smirl: Surprise minimizing reinforcement learning in unstable environments." arXiv preprint arXiv:1912.05510 (2019).
>
> **Q4) Related work to stabilize the intrinsic rewards**
>
> We thank the reviewer for suggesting we better contextualize our work with related previous works. As aforementioned and also suggested by other reviewers, we have now included a quantitative comparison between SOFE and DeRL.
>
> Even though there exist many monotonically decreasing functions of the state-visitation frequencies that can be used to compute count-based bonuses (e.g. like $\frac{1}{n}$), we argue that the two that we study in this work represent the most popular ones and that has been more thoroughly studied.
>
> Please see Section 3.2.1
>
> **Q5) Implementation of pseudo-counts**
>
> We thank the reviewer for raising this important point. We use the E3B algorithm to obtain an ellipsoid that represents the pseudo-state-visitation frequencies in a latent space, and not only the pseudo-count for a single state. By using SOFE, we augment the agents’ observations with the approximate state-visitation frequencies over the complete state space and show great performance gains. We have modified the paper to better express this very important point.
>
> Please see Section 3.2.2
>
> **Q6) Details on Figures**
>
> We thank the reviewer for raising this point. We have now added the necessary information for all figures.
>
> **Q7) SOTA baselines**
>
> E3B achieved SOTA results in procedurally generated environments. ICM and RND remain the most popular deep exploration algorithms. We thank the reviewer for mentioning these 2 works. While related, they do not provide open-source implementations of their methods. However, we have added them in the Related Work.
>
> We again thank the reviewer for the valuable feedback and invite him/her to re-evaluate the improved version of the paper. We have significantly improved the presentation, broadened the contribution, and provided a more sound empirical evaluation.

---

> > ### Author Response · Authors · 2023-11-20
> >
> > Dear Reviewer,
> >
> > We hope that you've had a chance to read our responses and clarifications. As the end of the discussion period is approaching, we would greatly appreciate it if you could confirm that our updates have addressed your concerns.
> >
> > Thank you very much.

---

> ### Author Response · Authors · 2023-11-21
>
> Dear Reviewer,
>
> The rebuttal time is almost over, and we would greatly appreciate it if you could confirm that our updates have addressed your concerns.
>
> Thank you very much.

---

> > ### Comment · Reviewer_Yao3 · 2023-11-21
> >
> > I want to thank the authors for their extremely detailed response and I am really sorry for my late reply.
> >
> > Their effort in changing the paper to accomodate reviewers' suggestions is truly remarkable. From a brief inspection, the paper seems to have improved substantially. I will consider increasing my score after a more detailed read of the new version.
> >
> > I do not have further questions for the authors. As a minor clarification, my point on state entropy maximization was that it is already a stationary objective, which does not need SOFE to become stationary.

---

> > > ### Author Response · Authors · 2023-11-21
> > >
> > > Thank you for your comments. We have now updated the last version of the paper.

---

> > > > ### Comment · Reviewer_Yao3 · 2023-12-03
> > > >
> > > > Having gone through the changes implemented in the manuscript, I am changing my score to 6.

---

### Official Review · Reviewer_4EvL · 2023-10-30

**Soundness:** 2 fair
**Presentation:** 2 fair
**Contribution:** 2 fair
**Rating:** 5
**Confidence:** 3

**Summary:**

The paper claims that exploration bonus induces a non-stationary reward function, which causes instability in policy learning. The paper proposes to solve this instability by incorporating a sufficient statistic over the exploration bonus. The paper conducts experiments on 2D and 3D maze-like environments and demonstrates the proposed method can cover the state space more efficiently when compared to existing approaches.

**Strengths:**

- The idea is simple and intuitive---by augmenting the state space the method converts the non-stationary exploration bonus to stationary.
- The results on navigation tasks seem promising---in particular the policy appears to explore the maze more efficiently and recovers behaviour akin to a goal-conditioned policy when the goal state is set to unvisited

**Weaknesses:**

**Comments**
I am happy to increase my score after these points are addressed:
- The formulation in E3B doesn't really cover the action which is also important for count-based exploration for particular tasks (e.g. deterministic dynamics will be okay but not stochastic). Consequently, I don't think this is particularly a sufficient statistic but only for environments tested.
	- This also raises another question---why does the paper only focus on navigation tasks? There are many other difficult exploration tasks (e.g. Montezuma's Revenge on Atari, Minecraft, etc.)
	- The paper claims that it is a sufficient statistic and empirically demonstrated that it does perform well, but it will be great if there is a proof showing that this is true under some assumptions.
- In section 4, the paper indicates that "we consider that the only unobserved components in the POMDP are the parameters of the reward distribution." I am curious as to why this is a good assumption? In particular if $\phi_t$ is only a sufficient statistic for exploration bonus, it may not be a sufficient statistic for deriving the state $s_t$ (with observation $o_t$).
	- By adding $\phi_t$ to the state space, we are essentially exploding the state space (potentially to $\infty$.) from a finite state space (e.g. the maze example in experimentation.) What is the intuition that the algorithm is still able to tractably find a optimal policy?
	- In continuous state-action space, the counting mechanism depends on the quantization---how do we still ensure stationary reward in this case? I believe it will be non-stationary reward since we cannot differentiate two states within the same bin. Otherwise, we can just use global timestep as the count.

**Questions:**

- I find the paper confusing to read at times:
	- There is some mention of $\phi_t$ but in experimentation we only mention $N_t$ (i.e. count) and $C_t$ (i.e. elliptical bonus). Are they the $\phi_t$?
	- Under section 5.1, what is $N_0$? Is it a binary mask over the map in the maze with only the $j$'th cell is 0? Is the observation the "state"?
	- What is salesman reward and $\sqrt{}$-reward on figure 5? I believe it is Eq. 3 and 2 respectively

**Possible typos**
- Abstract, fourth last line: "holds" instead of "hold"
- Generally, "state visitation frequency" should be "state-visitation frequency"
- Page 3, first paragraph, fourth last line: "should not" instead of "shouldn't"
- Five lines after Eq. 1: $\forall s_t, a_t$ instead of $\forall_{a, s, t}$
- Page 4, second paragraph, line 1: "Eq. 2 and 3" instead of "2 and 3".

---

> ### Author Response · Authors · 2023-11-13
> **Response to Reviewer 4EvL**
>
> We thank the reviewer for the relevant feedback. In line with your comments, we have clarified several key ideas in the paper, improved the presentation, and broadened the contribution.
>
> **1) Formulation of E3B**
>
> Regarding the formulation in E3B, our choice aligns with the original implementation (and with the typical literature in exploration bonuses), which measures novelty over the state space. While count-based methods can be defined over state-action pairs, many popular novelty-seeking objectives, including RND, ICM, E3B, and CoinFlip Networks, aim to maximize exploration over the state space.
>
> See Page 2, Column 2, Paragraph 1, last sentence [here](https://arxiv.org/abs/2306.03186)
>
> **1.1) Environments used and navigation**
>
> Minihack, Procgen, and Habitat are widely recognized benchmarks for evaluating exploration in deep RL. Both Minihack and Habitat were intentionally selected because they allow for a direct comparison with existing methods (e.g. E3B). While Montezuma could be defined as a navigation environment, it has other mechanics that go beyond conventional navigation tasks  (e.g. avoiding enemies/climbing ladders). Similarly, in our experiments in Section 5.1, we introduce a 3D world with challenges analogous to Montezuma. Here, the agent must efficiently use jump pads and avoid lava and water pools while still aiming to cover the entire map successfully. Additionally, we designed the 3D world to mirror the complexity of open-world navigation seen in Minecraft. Minecraft has large computing requirements, and it takes more than a week to train an agent in Montezuma (see [here](https://docs.cleanrl.dev/rl-algorithms/ppo-rnd/#implementation-details)) (2 billion samples) and is often not included in papers for this reason.  With this, we believe that our proposed mazes, together with DeepSea, the 3D world, Minihack, Procgen-Maze and Habitat, are representative of the hard-exploration settings widely studied in the literature of intrinsic rewards for exploration in deep RL.
>
> Please see lines 284-289.
>
> **1.2) Solving non-stationarity and identifying sufficient statistics**
>
> Regarding our claims on solving the non-stationarity, we have modified the paper to better express that SOFE augments the observations with all the moving components of the intrinsic rewards. The state-visitation frequencies are the only dynamically changing components in Equations 1 and 2, and so are the ellipsoid matrix in Equation 3, and the generative model in Equation 5. We have modified the paper to better explain this point.
>
> Please see lines 161-163, and 179-182.
>
> An example to see that the state-visitation frequencies are sufficient statistics that allow for the existence of an optimal Markovian stationary policy is the following: Consider an MDP with paths S1 → S2 → S3 and S3 → S2 → S1 (no direct path between S3 and S1). The agent always starts at S1. To optimize for the count-based reward $R(s) = \frac{1}{N(s)}$ the optimal policy is to go S1→S2→S3→S2→S1→S2→S3 – but of course, this is not a stationary policy (needs to swap actions in S2 depending on the counts of S1 and S3). The best stationary policy here is S1→S2, S3→S2 and then a 50/50 chance from S2→S1 and S2→S3 – but this is suboptimal. The optimal policy described above can only be learned with access to the state-visitation frequencies (provided by SOFE):  if the agent observed the counts, then the optimal policy is stationary: at S2, if N(S3) > N(S1) go to S1, else go to S3.
>
> **2) Assumptions on the unobserved components of the states**
>
> This assumption holds true in many cases.  In particular, it holds in the maze environments used in the paper, or any grid world environment in general, where the agent has access to sufficient statistics of the transition dynamics. Even in Atari environments like Breakout, where the assumption might not hold because a stack of observations is required to infer the true state of the game (e.g. direction of the bullets), SOFE mitigates the non-stationary optimization, yielding performance gains. Importantly, this is evident in our experiments on the 3D world, Habitat, Procgen-Maze, and Minihack, where the agent has only a partially observable view of the environment and yet benefits from SOFE.
>
> **3) By adding $\phi_t$ to the state space, we are essentially exploding the state space (potentially to $infty\$**
>
> While it is true that we are augmenting the state space, we add features that provide information about the hidden dynamics of the rewards and help improve policy convergence. As you note, there are more subtle interactions occurring between the size of the state space and training dynamics, and thanks to deep learning, increasing the size of the state space is less problematic for learning than having non-stationary objectives.
>
> Please see lines 49-54.
>
> In the next comment, we tackle the remaining comments.

---

> > ### Comment · Reviewer_4EvL · 2023-11-17
> >
> > **W1:**
> > - I agree that many count-based methods consider only state spaces. Just a question about this field generally and I welcome discussions---why we should only consider state spaces generally? It seems like considering only state spaces remove the notion of controllability of the agent itself.
> >
> > **W1.1:**
> > - Thank you for addressing this concern.
> >
> > **W1.2:**
> > - Thank you for providing an example. In this case, it is true that we simply need the state-visitation frequencies, but this seems to be suitable for a particular MDP structure? Can you clarify what kind of MDPs should we be expecting for this method to work well?
> >
> > **W2:**
> > - Thank you for explaining the limitation of this assumption. I think the paper should include this so readers can consider the situation for which they should use this approach.
> >
> > **W3:**
> > - Thank you for providing a justification. I think these subtle interactions need to be explained more---while I also leaned towards "thanks to deep learning" we empirical get good results, I would really love to see more convincing arguments. Even just a linear MDP as a sanity check will be sufficient.

---

> > > ### Author Response · Authors · 2023-11-20
> > >
> > > We thank the reviewer for their responses. We now address the remaining questions:
> > >
> > > **W1) Why we should only consider state spaces generally? It seems like considering only state spaces remove the notion of controllability of the agent itself.**
> > >
> > > We acknowledge that exploratory agents should maximize novelty in the joint state-action space, as the task rewards we are interested in finding are also a function of the state-action pairs. It is likely new methods will come up in the future that will achieve better performance in hard-exploration tasks by optimizing for this joint objective. However, current methods still define novelty purely in the state space (e.g. state-prediction error, state-entropy maximization), and our work is about improving current exploration objectives. By proposing SOFE, we expect future research works on exploration objectives that are broader (e.g. take into account states and actions) to be careful about the implicit non-stationarity of their approaches and resolve it with a similar state-augmentation as the ones we propose in our current work.
> > >
> > > **W1.2) It is true that we simply need the state-visitation frequencies, but this seems to be suitable for a particular MDP structure? Can you clarify what kind of MDPs should we be expecting for this method to work well?**
> > >
> > > We thank the reviewer for raising this point. This is an important point because SOFE is not tied to a specific MDP structure. It is crucial to note that any non-stationary reward distribution induces a partially observed MDP and generally removes the possibility for a Markovian policy to be optimal. Our proposed solution to obtain an optimal Markovian policy in this case is again to augment the states with sufficient statistics of the non-stationary rewards. Consider a more general example that is not related to count-based bonuses, as mentioned in the paper (see lines 137-140):
> > >
> > > "Consider an MDP where the reward distribution is different at odd and even time steps. If the states of the MDP are not augmented with an odd/even component (i.e., in this case, being sufficient statistics of the rewards), the rewards appear to be non-stationary to an agent with a Markovian policy. In this case, a Markovian policy will not be optimal over all policies. The optimal policy will have to switch at odd/even time steps and hence require the augmented information."
> > >
> > > This example is not related to count-based rewards but shows that in general, a non-stationary reward distribution makes Markovian policies suboptimal, but can be fixed by providing the necessary information to the agents. In this work, we extrapolate this argument to intrinsic exploration objectives, which induce non-stationary rewards. We have added lines 141-143 to make this point more clear.
> > >
> > > **W2) "I think the paper should include this so readers can consider the situation for which they should use this approach"**
> > >
> > > We have added information about this discussion in the updated version of the paper (see footnote 3 in Section 4).
> > >
> > > Footnote 3: "This assumption holds true if the agent has access to sufficient statistics of the transition dynamics (e.g.
> > > grid environments), and makes SOFE transform a POMDP into a fully observed MDP. Even when there are
> > > unobserved components of the true states apart from the parameters of the intrinsic reward distributions, we
> > > show that SOFE mitigates the non-stationary optimization, yielding performance gains"
> > >
> > > **W3)  I think these subtle interactions need to be explained more---while I also leaned towards "thanks to deep learning" we empirical get good results, I would really love to see more convincing arguments. Even just a linear MDP as a sanity check will be sufficient.**)
> > >
> > > We thank the reviewer for following up on this discussion. In the following, we argue why SOFE achieves such significant performance gains in many hard-exploration tasks and can converge faster to optimal policies:
> > >
> > > Throughout the paper, we study global and episodic exploration, and we consider different reward distributions (e.g. Equations 1 and 2). In episodic exploration, the objective is to cover the state space in a single finite-length episode, and the state-visitation frequencies are reset at the beginning of each episode. For episodes of length T, any state can be visited at most T times, and hence the augmented state space with SOFE is bounded and remains finite. Hence, the augmented MDP allows for the same proof of convergence to an optimal Markovian stationary policy in Chapter 5 in [1]. Even in global exploration, when using the reward distribution in Equation 2, the proof remains valid as the augmented state space remains finite.
> > >
> > > [1] Puterman, Martin L. Markov decision processes: discrete stochastic dynamic programming. John Wiley & Sons, 2014.
> > >
> > > This argument shows that SOFE augments the state space, but the modified optimization problem remains tractable.
> > >
> > > In the following, we address your remaining comments.

---

> > > > ### Comment · Reviewer_4EvL · 2023-11-21
> > > >
> > > > Thank you for the response.
> > > >
> > > > **W1.2:** Generally, my understanding is that you can augment any (PO)MDPs such that you end up having an optimal stationary Markovian policy. I believe what is confusing to me is that SOFE ends up using only state-next-state pairs for computing "sufficient statistics" which I interpret to be the information required to convert POMDPs into MDPs. However, as you have mentioned in **W1** that state-next-state pairs are potentially insufficient in all MDPs.
> > > >
> > > > **W2:** Thank you for including the footnote. At the end of the foot note, I believe it is better to indicate "we empirically show that SOFE mitigates ..."
> > > >
> > > > **W3:** Can you clarify how this deals with the global exploration setting? We assume the augmented state space to be only up to the maximum number of allowed interaction with the environment? How meaningful is this bound when $T$ is absorbed by the state space which may be significantly higher than the original state space $\mathcal{S}$?

---

> > > > > ### Author Response · Authors · 2023-11-21
> > > > >
> > > > > Thank you again for helping us improve our work.
> > > > >
> > > > > **W1.2) Generally, my understanding is that you can augment any (PO)MDPs such that you end up having an optimal stationary Markovian policy. I believe what is confusing to me is that SOFE ends up using only state-next-state pairs for computing "sufficient statistics" which I interpret to be the information required to convert POMDPs into MDPs. However, as you have mentioned in W1 that state-next-state pairs are potentially insufficient in all MDPs.**
> > > > >
> > > > > Importantly, in the paper, we argue that the state-next-state transitions are enough to update the sufficient statistics of the reward distributions, and this reflects the Markovian property. Concretely, the state-visitation frequencies (i.e. counts), the E3B ellipsoid, and the generative model of S-Max obey Markovian dynamics in their updates, as we show in lines 225-231.
> > > > >
> > > > > Even in the aforementioned case where there might remain unobserved components of the states, and SOFE does not fully transform the POMDP into a fully-observed MDP, our experiments show that by solving the non-stationarity of the rewards with SOFE, agents explore the state space significantly better (as discussed in the previously suggested footnote).
> > > > >
> > > > > **W2) Thank you for including the footnote. At the end of the foot note, I believe it is better to indicate "we empirically show that SOFE mitigates ..."**
> > > > >
> > > > > We have modified the footnote to make this point more clear.
> > > > >
> > > > > **W3) Can you clarify how this deals with the global exploration setting? We assume the augmented state space to be only up to the maximum number of allowed interaction with the environment? How meaningful is this bound when T is absorbed by the state space which may be significantly higher than the original state space S?**
> > > > >
> > > > > Importantly, in the global settings, the state space is much bigger as you mention, since it is bounded to the maximum number of allowed environment interactions. However, note that not all states in the augmented MDP are visitable, as once a state $s_t$ is visited, all the possible augmented combinatorial states that had associated state-visitation frequencies where $s_t$ had not been visited are not valid anymore, and hence the augmented state space is continually truncated as the agent interacts with the environment.  Furthermore, since the updates on the state-visitation frequencies follow Markovian dynamics and are smooth, the agents augmented with SOFE can easily learn representations that generalize across different state-visitation frequencies (as we show in Figure 10, SOFE agents can leverage efficiently state-visitation frequencies that the agent had not seen during training).
> > > > >
> > > > > One again, we thank the reviewer for their valuable feedback.

---

> > > > > ### Author Response · Authors · 2023-11-22
> > > > >
> > > > > Dear Reviewer,
> > > > >
> > > > > We hope that you've had a chance to read our responses to your remaining concerns. As the end of the discussion period is in less than 24 hours, we would greatly appreciate it if you could confirm that our updates have addressed your concerns, and we invite you to re-evaluate the paper if that is the case.
> > > > >
> > > > > Thank you very much.

---

> > > > > > ### Comment · Reviewer_4EvL · 2023-11-22
> > > > > >
> > > > > > Thank you for the response. I don't think I am totally convinced and would also invite other reviewers to help me understand why and how **W1.2** and **W3** are addressed. I will keep my score for now, thank you again for the discussions.

---

> > > > > > > ### Author Response · Authors · 2023-11-23
> > > > > > >
> > > > > > > Thank you for your continued involvement in the discussion, it is greatly appreciated. If you can highlight what about our previous answer was unclear we can provide a more detailed response. However, we provide more details below.
> > > > > > >
> > > > > > > **W1.2)** We can apply SOFE to any MDP as it is possible to capture or estimate the visitation distribution of any MDP.
> > > > > > >
> > > > > > > **"I believe what is confusing to me is that SOFE ends up using only state-next-state pairs for computing "sufficient statistics" which I interpret to be the information required to convert POMDPs into MDPs."**
> > > > > > >
> > > > > > > The sufficient statistics that we use in SOFE are of the **visitation distribution**, and the latter is a Markovian function (the function at the next state only depends on the previous state and action of the agent, and that is why the state-next-state transitions are sufficient). Independently of the structure of the POMDP, we are able to make the intrinsic reward distributions stationary with sufficient statistics of the visitation distribution and that is the objective of our paper. If the POMDP contains other unobserved components, it is not the objective of SOFE to estimate them, as in the paper we show that just by making the rewards stationary, deep RL agents achieve significant performance gains.
> > > > > > >
> > > > > > > W3: Prior to adding $\phi_t$, it is unlikely that we can train a policy to converge because of the shifting reward function. While SOFE adds features and expands the state space, they are the features to determine the reward function and result in a stationary training distribution. Importantly, in the paper, we empirically show that in deep RL, agents are better at generalizing across large state spaces than at optimizing non-stationary rewards. In fact, deep RL has achieved great achievements in MDPs with large state spaces -- any pixel-based environment has a huge state space and still can be solved by leveraging the generalization capabilities of neural networks. With this, we put or focus on the optimization challenges of non-stationary rewards and, as all other deep RL works do, we leverage deep networks to generalize across the large state space.

---

> ### Author Response · Authors · 2023-11-14
>
> **4) Discretizing a continuous state space and non-stationarity**
>
> By discretizing the continuous state space into bins, we define an auxiliary MDP, instantiating the exploration problem within this discrete framework. Since the exploration objective, observations, and state-visitation frequencies are computed over this auxiliary state space, the true continuous space remains untouched and hence does not induce any non-stationarity on the reward function in the auxiliary MDP.
>
> Regarding the use of the time step $t$ to augment the states, we argue that it does not have an effect on the non-stationarity of the reward distribution. However, it does affect the stationarity of the transition function. Moreover, it has been shown in [1] that including the time step information in the states can lead to degenerate policies. Please see our discussion with the Reviewer 7vAW in **Q1**
>
> [1] Pardo, Fabio, et al. "Time limits in reinforcement learning." International Conference on Machine Learning. PMLR, 2018.
>
> **Q1) There is some mention of $\phi_t$ but in experimentation we only mention $N_t$  (i.e. count) and $C_t$ (i.e. elliptical bonus). Are they the $\phi_t$?**
>
> Correct, SOFE is a general framework applicable to any sufficient statistics $\phi_t$. In this paper, we apply SOFE to count-based methods, E3B,  and state-entropy maximization. We have modified the paper to make the notation more clear.
>
> Please see the improved Section 3.2.
>
> **Q2) What is $N_0$ in Section 5.1**
>
> Correct,  it is a binary mask over the map in the maze with only the j'th cell as 0 and all the other cells as 1's. For this analysis, we design $N_0$, marking all states as visited except for a single one. The SOFE agent, when provided with the maze observation and binary mask, has learned to pay attention to the minutest details in $N_t$, and directs exploration efficiently toward the single unvisited state.
>
> **Q3) What is salesman reward and $\sqrt{}$-reward on figure 5**
>
> The $\sqrt{}$ and salesman rewards are shown in Equations 1 and 2, respectively. To enhance clarity, we've added a sentence to explicitly connect these reward formulations.
>
> Please see Section 3.2.1
>
> We again thank the reviewer for the valuable feedback and we believe we have addressed all concerns from the reviewer.

---

> ### Comment · Reviewer_4EvL · 2023-11-17
>
> Thank you for the detailed response.
>
> **W4:**
> - So if I understand correctly, we are considering the exploration of the discretized MDP, not the underlying continuous MDP. If that is the case, I am wondering how should we effectively explore the actual continuous MDP using this method? Or is this a limitation currently?
>
> - Right, the global timestep in a sense is captured by the counts. This was raised since I was not sure whether the exploration over the true continuous MDP is being evaluated. If not, this sounds reasonable to me.
>
> Thanks for clarifying the remaining questions.
>
> EDIT: I just noticed the ordering is flipped and I apologize for that. Nevertheless I have increased the score from 3 to 5 due to remaining questions.

---

> > ### Author Response · Authors · 2023-11-20
> >
> > **W4)  If I understand correctly, we are considering the exploration of the discretized MDP, not the underlying continuous MDP. If that is the case, I am wondering how should we effectively explore the actual continuous MDP using this method? Or is this a limitation currently?**
> >
> > Correct, in the experiments shown in Figure 4 we discretize the observation space and instantiate the count-based rewards in the discretized MDP. This was done to further evaluate the performance of SOFE in cases where the state-visitation frequencies are estimates of the true (and hidden) states of the environment. This is indeed a limitation of count-based rewards - not of SOFE - which require the state space to be discrete and countable.
> >
> > To better estimate the state-visitation frequencies in continuous and high-dimensional spaces, recent works like E3B open the path for computing novelty directly in continuous spaces. However, they require learning a density (or neural) model of the visitation frequencies. We hypothesize that in the 3D world used for our experiments in Figure 4, E3B would explore better the state space than not the discretized counts. However, this comparison lies out of the scope of our work, as importantly, we show that SOFE provides orthogonal gains to different exploration objectives of different natures, independently of their capabilities and limitations. We hope this clarifies that SOFE does not inherit the limitations of any particular exploration objective, but improves the optimization of many different objectives independently of their specifications.
> >
> > Once again, we appreciate your constructive feedback.

---

### Official Review · Reviewer_7vAW · 2023-11-01

**Soundness:** 3 good
**Presentation:** 3 good
**Contribution:** 3 good
**Rating:** 8
**Confidence:** 4

**Summary:**

This paper focuses on a problem faced by approaches that use intrinsic rewards for guiding exploration; they introduce non-stationarity in the reinforcement learning objective. This non-stationarity in rewards can destabilize learning in many RL approaches. The non-stationarity arises because the agent cannot predict the intrinsic reward as the features required to predict it (such as visitation counts) are not observable.

Focussing on generalizations of count-based approaches, the paper proposes Stationary Objectives For Exploration (SOFE) that aim to augment the state space with sufficient statistics for intrinsic reward prediction to eliminate the partial observability and associated non-stationarity.

Experiments in sparse and no-reward navigation tasks show that including SOFE improves the performance of a previously proposed count-based intrinsic reward approach (E3B).

**Strengths:**

**S1.** Using count-based bonuses to explore sparse-reward environments is a widespread technique in RL. Improving the performance of count-based approaches would interest the research community.

**S2.** The paper focuses on a relatively under-explored issue of addressing the non-stationarity introduced due to count-based bonuses. The proposed solution of augmenting states with sufficient statistics for intrinsic rewards seems novel, simple, and well-motivated.

**S3.** The paper presents concepts clearly and is easy to follow.

**Weaknesses:**

**W1.** The paper should adjust the claim that it identifies that intrinsic reward functions induce a non-stationary RL objective. While solutions may be under-explored, the issue is generally known and noted in previous papers (e.g., [1]). Other works have chosen not to tackle the non-stationarity in intrinsic reward as it slowly varies and could be tracked [2]. Also, see the references in W2.

**W2.** A crucial weakness of the paper is that it misses comparisons with previous works that have proposed decoupling exploration and exploitation policies [3, 4] to tackle the non-stationarity introduced due to intrinsic bonuses. The paper would significantly benefit from comparing their approach with baselines based on decoupling exploration and exploitation.

**W3.** An aspect that merits further discussion is that RL algorithms with recurrent architectures (like PPO + LSTM used in the paper) could learn representations that resolve this partial observability [5]. One way to promote this effect could be to have an auxiliary task of predicting intrinsic rewards (which could make for an interesting baseline).

While this might perform worse compared to directly providing sufficient statistics in the state (in terms of sample efficiency), it is potentially a more general solution to the problem for other intrinsic rewards.
For instance, it has previously been shown that LSTMs can learn to count in discrete settings [6], which could be helpful in episodic exploration settings. Previous work has also shown that recurrent architectures can learn contexts that resolve non-stationarity due to partial observability in bandit problems [7].


Overall, I appreciate the direction the authors took to address the non-stationarity introduced by intrinsic rewards. Should the weaknesses and questions be adequately addressed/clarified, I would gladly increase my score.


—------------------—------------------—------------------—------------------—------------------

### References

[1]Singh, S., Lewis, R. L., Barto, A. G., & Sorg, J. (2010). Intrinsically motivated reinforcement learning: An evolutionary perspective. IEEE Transactions on Autonomous Mental Development, 2(2), 70-82.

[2] Şimşek, Ö., & Barto, A. G. (2006, June). An intrinsic reward mechanism for efficient exploration. In Proceedings of the 23rd international conference on Machine learning (pp. 833-840).

[3] Whitney, W. F., Bloesch, M., Springenberg, J. T., Abdolmaleki, A., Cho, K., & Riedmiller, M. (2021). Decoupled exploration and exploitation policies for sample-efficient reinforcement learning. arXiv preprint arXiv:2101.09458.

[4] Schäfer, L., Christianos, F., Hanna, J. P., & Albrecht, S. V. (2021). Decoupled reinforcement learning to stabilise intrinsically-motivated exploration. arXiv preprint arXiv:2107.08966.

[5] Ni, T., Eysenbach, B., & Salakhutdinov, R. (2021). Recurrent model-free rl can be a strong baseline for many pomdps. arXiv preprint arXiv:2110.05038.

[6] Suzgun, M., Belinkov, Y., & Shieber, S. M. (2019). On Evaluating the Generalization of LSTM Models in Formal Languages. In Proceedings of the Society for Computation in Linguistics (SCiL)

[7] Ramesh, A., Rauber, P., Conserva, M., & Schmidhuber, J. (2022). Recurrent Neural-Linear Posterior Sampling for Nonstationary Contextual Bandits. Neural Computation, 34

**Questions:**

Q1. There still remains a source of non-stationarity for RL as the considered tasks have a maximum number of steps, but agents are not provided the time step in the state. In the episodic exploration setting $C_t$ can probably be used to infer the time step. However, it might make sense to include the time step in the global exploration setting. I am curious to know the authors’ thoughts regarding this and whether they have tried something along these lines.

Q2. Is there a reason why Figure 3 shows visitations with A2C as the base algorithm, but the evaluation in Figure 4 uses PPO? Would these qualitative results change with the base algorithm?

---

> ### Author Response · Authors · 2023-11-13
> **Response to Reviewer 7vAW**
>
> We thank the reviewer for the insightful comments. We have addressed your points and improved the paper. We now better highlight the generality of SOFE, providing results across multiple exploration objectives. Importantly, and in line with your comments, we have adjusted the claims of our paper, and better positioned our work with respect to previous related research. We have also included relevant experiments that allow for the necessary comparisons to previous methods, as you rightfully suggested. We have significantly improved the presentation, broadened the contribution, and provided a more sound empirical evaluation.
>
> **1) Adjusting the claims about the non-stationary intrinsic rewards**
> We acknowledge the reviewer's observation and have adjusted our claim in the paper. The modifications emphasize the non-stationarity of intrinsic rewards as a recognized issue, drawing connections to observations from various previous works. The core motivation for our work remains to provide a novel solution to address this challenge.
>
> Please see lines 4-5, 40-48, 103-106 and 109.
>
> **2)  Comparisons with related work that decouples exploration and exploitation**
> We appreciate the reviewer's observation and have taken steps to address this concern. We have run experiments on the DeepSea environment (as used in the DeRL paper [1]). Results indicate that SOFE, which simplifies the learning process by training a single policy end-to-end with stationary intrinsic and extrinsic rewards, outperforms DeRL in the harder exploration variations of the environment (see Table 1 in the paper).  We again thank the reviewer since these new results provide further evidence which strengthens our paper by showing the superior performance of SOFE in challenging exploration tasks.
>
> Please see lines 40-44, 103-106, 256-263, 310-314, and Table 1.
>
> [1] Schäfer, L., Christianos, F., Hanna, J. P., & Albrecht, S. V. (2021). Decoupled reinforcement learning to stabilise intrinsically-motivated exploration. arXiv preprint arXiv:2107.08966.
>
> **3) Recurrent architectures**
>
> Thank you for highlighting the importance of clarifying our choice of the PPO+LSTM baseline. In line with your comments, our rationale for including this architecture is to investigate whether it can effectively compensate for the non-stationary rewards. We have modified the paper to better motivate this analysis.
>
> Please see lines 306-309.
>
> Regarding the training process of the recurrent policy, the LSTM extracts features from observations, which are then passed to the critic, aiming to regress the episodic return (in this case, of intrinsic rewards). This procedure already enables learning trajectory representations that compensate for the dynamically changing rewards, similar to the intrinsic reward prediction auxiliary task that you mention.
>
> Indeed, our results show that agents equipped with an LSTM generally perform better than those using simpler algorithms like vanilla DQN and A2C, but SOFE still provides performance gains on top of this higher-capacity architecture (see Figure 7).
>
> **Q1) Time limit and non-stationarity**
>
> We thank the reviewer for raising this point. We believe that once the sufficient statistics of the intrinsic rewards are observed by the agent the reward distributions become fully Markovian, and hence do not require the time step. Note that in Equations 1-2-3-5 the reward at time step t+1 only depends on the sufficient statistics at time t. Since this reflects the Markovian property, it is invariant across t.  We have modified the paper to better explain this point.
>
> Please see lines 215-218.
>
> Importantly, we note that not including the time step in the observations can induce non-stationary transition dynamics as shown in [1]. However, [1] shows that including it can lead to degenerate policies, and not including it communicates to the agent that there is a non-zero probability of the episode terminating at each step, which helps learn more consistent policies. Still, in our work, we aim to make the reward function stationary, and as argued above, this is already achieved without including the timestep information.
>
> [1] Pardo, Fabio, et al. "Time limits in reinforcement learning." International Conference on Machine Learning. PMLR, 2018.
>
> **Q2) A2C vs PPO in the Figures**
>
> Thank you for raising this point. We want to clarify that the choice of A2C in Figure 3 and PPO in Figure 4 was made to illustrate that SOFE is agnostic to the RL algorithm. The qualitative results are consistent across these algorithms. To enhance clarity, we are open to fixing a single algorithm for these results. Furthermore, the results in Figure 3 correspond to the curves shown in Appendix Section A.9.2. The results for all other algorithms are also shown in the Appendix and they show that SOFE is generally beneficial.
>
> We again thank the reviewer for the valuable feedback and we believe we have addressed all concerns from the reviewer.

---

> > ### Comment · Reviewer_7vAW · 2023-11-18
> >
> > Thank you for your detailed response. It addresses most of my concerns. I greatly appreciate the time and effort spent revising the paper and conducting the additional experiments. Before updating my score, I would like to clarify one issue with the new experiment (regarding W2).
> >
> > **Q3.** Decoupled exploration and exploitation: In the DeepSea experiments, why does DERL-DQN improve when depth is increased from 20 to 24? That seems a bit strange. Are these results with multiple initial seeds for the RL agents?
> >
> > On a minor note, regarding the comment about recurrent architectures (3), I agree that there is pressure on the RNN to resolve partial observability due to the objective of the critic. However, predicting returns mixes information across timesteps compared to an auxiliary task of predicting (intrinsic) rewards to resolve non-stationarity (like in [7] or other meta-RNN-based RL approaches). The extra features provided in the case of SOFE allow for the prediction of intrinsic rewards, which is a stronger condition than having to predict ‘intrinsic returns’. Nevertheless, I don’t believe such an experiment is a necessity here as it can be considered beyond the scope of this paper.

---

> > > ### Author Response · Authors · 2023-11-19
> > >
> > > Thank you for your responses.
> > >
> > > **Q3) "Why does DERL-DQN improve when depth is increased from 20 to 24? That seems a bit strange. Are these results with multiple initial seeds for the RL agents?"**
> > >
> > > For DeRL, we have reproduced the results published in the original paper (refer to Section 5.4 in [1], Table 1). The reported results for DeRL in DeepSea were obtained by averaging across 5 random seeds and using the best hyperparameter configuration identified through extensive tuning experiments. Similarly, the results for SOFE follow the same evaluation protocol, averaging the returns across 5 seeds and training for 100k episodes on each variation of the environment. To enhance clarity, we have included this information in the Table caption.
> > >
> > > [1] Schäfer, Lukas, et al. "Decoupled reinforcement learning to stabilise intrinsically-motivated exploration." arXiv preprint arXiv:2107.08966 (2021).
> > >
> > > Regarding the recurrent architectures: We acknowledge your point, and although we maintain that this comparison falls outside the paper's scope, we recognize the potential of exploring auxiliary tasks to learn representations capable of addressing the non-stationarity of intrinsic rewards. This idea is indeed promising and merits further investigation.
> > >
> > > Once again, we appreciate your constructive feedback.

---

> > > > ### Comment · Reviewer_7vAW · 2023-11-20
> > > >
> > > > Thanks for the clarifications. As mentioned earlier, the response addresses most of my concerns. I have updated my score accordingly.

---

### Meta-Review · Area_Chair_LVuz · 2023-12-05

**Metareview:**

This paper proposes a solution to the non-stationarity induced by intrinsic rewards used to induce exploration in reinforcement learning. The paper proposes the use of auxiliary inputs in order to eliminate the partial observability and associated non-stationarity of such a problem. I am recommending the paper to be accepted as it discusses a relevant topic for exploration in reinforcement learning and proposes a simple and effective solution to the issues they raised. The authors did a fairly good job addressing the concerns raised by the reviewers during the discussion phase.

**Justification For Why Not Higher Score:**

The paper is very borderline, given how much controversy it has generated, I don't see why it could get a higher score than it just being accepted.

**Justification For Why Not Lower Score:**

I struggled with this one a little bit. It does feel the paper is above the bar, and the reviewers pushing back are maybe inexperienced and they are just holding the paper to weird standards. That being said, I definitely wouldn't be upset if this paper were to be bumped down.

---

### Decision · Program_Chairs · 2024-01-16

Accept (poster)